# Online Learning with Recency:
# Algorithms for Sliding-window Streaming Multi-armed Bandits

Vladimir Braverman [1] [2]   Chen Wang [3]   Liudeng Wang [4]   Samson Zhou [4]

## Abstract

Motivated by the recency effect in online learning, we study algorithms for single-pass *sliding-window streaming multi-armed bandits (MABs)* in this paper. In this setting, we are given $n$ arms with unknown sub-Gaussian reward distributions and a parameter $W$. The arms arrive in a single-pass stream, and only the most recent $W$ arms are considered valid. The algorithm is required to perform pure exploration and regret minimization with *limited memory*, defined as the number of stored arms. The model is a natural extension of the streaming multi-armed bandits model (without the sliding window) that has been extensively studied in recent years. We provide a comprehensive analysis of both the pure exploration and regret minimization problems with the model. For pure exploration, we prove that finding the best arm is hard with sublinear memory while finding an *approximate* best arm admits an efficient algorithm. For regret minimization, we explore a new notion of regret and give sharp memory-regret trade-offs for any single-pass algorithm. We complement our theoretical results with experiments, demonstrating the trade-offs between sample, regret, and memory.

## 1 Introduction

The stochastic multi-armed bandits (MABs) model is a fundamental model extensively studied in machine learning (ML) and theoretical computer science (TCS). In its most common form, we are given $n$ arms with unknown sub-Gaussian reward distributions, and we can learn the instance by *sampling* from the arms. The most important problems in the model include *pure exploration*, where the goal is to identify the best or a near-optimal arm, and *regret minimization*, where the aim is to devise a sampling strategy that performs competitively against the best arm in hindsight. The multi-armed bandits model has found broad applications in experiment design and clinical trials (Robbins, 1952; Pallmann et al., 2018; Simchi-Levi & Wang, 2023), financial strategies (Shen et al., 2015; Trovò et al., 2018), information retrieval (Radlinski et al., 2008; Losada et al., 2017), algorithm design (Bouneffouf et al., 2017; Gullo et al., 2023), to name a few.

Classical algorithms for MABs often assume the entire set of $n$ arms is stored in the memory for repeated access. However, this assumption can be unrealistic in modern online learning and large-scale applications, where arms may arrive sequentially in a stream, and the available memory is insufficient to store all of them. To address this challenge, the work of (Liau et al., 2018; Assadi & Wang, 2020) introduced the *streaming* multi-armed bandits model. In this model, the arms arrive one after another in a stream, and the algorithm would ideally maintain a memory substantially smaller than the total number of arms. The maximum number of arms maintained in the memory is defined as the *space complexity* of the algorithm. The streaming MABs model has attracted considerable attention, and a flurry of work has established near-tight trade-offs for pure exploration (Assadi & Wang, 2020; Jin et al., 2021; Maiti et al., 2021; Assadi & Wang, 2022; 2024; Karpov & Wang, 2025) and regret minimization (Liau et al., 2018; Maiti et al., 2021; Agarwal et al., 2022; Wang, 2023; He et al., 2025) in various settings.

While most work on streaming MABs targets global objectives, e.g., identifying the best arm in the entire stream, many applications exhibit a recency effect (Braverman et al., 2019), where recent arms matter more. For example, movie recommendation systems must adapt quickly to shifting trends (Bogunovic et al., 2017; Avdiukhin et al., 2019). Another motivation of the recency effect comes from privacy constraints: regulations and policies often mandate data deletion after limited periods. GDPR requires data retention only for the "necessary" duration (GDPR, 2016), Apple retains user data for 6 months (Apple Inc., 2021), and Google

[1]Johns Hopkins University [2]Google Research [3]Rensselaer Polytechnic Institute [4]Texas A&M University. Correspondence to: Vladimir Braverman <vova@cs.jhu.edu>, Chen Wang <wangc33@rpi.edu>, Liudeng Wang <eureka@tamu.edu>, Samson Zhou <samsonzhou@gmail.com>.

*Proceedings of the 43rd International Conference on Machine Learning*, Seoul, South Korea. PMLR 306, 2026. Copyright 2026 by the author(s).

limits anonymized advertising data to 9 months (Google LLC, 2025). Unfortunately, streaming MABs algorithms usually do not take any recency effect into consideration. For instance, the pure exploration algorithms, e.g., the ones in (Assadi & Wang, 2020; Jin et al., 2021; Maiti et al., 2021), may output an arm that arrives very early in the stream, which is far from being recent. Similarly, the regret minimization algorithms in (Maiti et al., 2021; Wang, 2023; He et al., 2025) may commit to an arm that is outside the pool of recent arms[1]. As such, we ask the following question: *can we design efficient streaming MABs algorithms that incorporate the recency effect?*

**Sliding-window streaming multi-armed bandits.** One of the most common models that capture the recency effect is the sliding-window streaming model (Datar et al., 2002; Datar & Motwani, 2016). In a typical sliding-window stream, a total of $n$ data items (arms in the context of MABs) are arriving in a stream, and only the past $W$ items are considered valid. The sliding-window streams have been extensively studied in various contexts, including frequency estimation (Datar et al., 2002; Braverman & Ostrovsky, 2007; Braverman et al., 2018; 2021; Woodruff & Zhou, 2021; Blocki et al., 2023; Feng et al., 2025; Nagawanshi et al., 2026), graph algorithms (Crouch et al., 2013; Crouch & Stubbs, 2014; Zhang et al., 2024), clustering (Braverman et al., 2016; Borassi et al., 2020; Epasto et al., 2022; Woodruff et al., 2023; Cohen-Addad et al., 2025), among others (Tao & Papadias, 2006; Zhang et al., 2016; Jayaram et al., 2022; Braverman et al., 2026).

Inspired by the success of sliding-window streams on various problems, we define the natural notion of sliding-window streaming MABs to explore the recency effect. Here, we are given $n$ arms arriving in a (single-pass) stream, and we are additionally given a window size $W$. When the $t$-th arm arrives, the arms with the arrival orders in $[t - W + 1, t]$ are considered the *valid* set of arms at this point. We emphasize that throughout the paper, $W$ and $n$ are parameters given by the problem instance, and we cannot adjust these parameters. The algorithm is only allowed to store *valid arms*, and any invalid arm is evicted from the memory immediately [2]. An illustration of the model can be found in Figure 1.

**Separation of data arrival and arm sampling.** A critical distinction in our model compared to existing MABs models is the separation between the data arrival (environment change) and the arm pulls. Some notable examples to

---

[1]This intuitively means the algorithm incurs large regret, although the definition of regret has more nuance in such cases. See Section 1.1 and Section 2 for details.

[2]In the standard sliding-window streaming setting, the algorithm is allowed to store expired items. A simple reduction to the standard sliding-window case can be done by *not* allowing any arm pull on the expired arms.

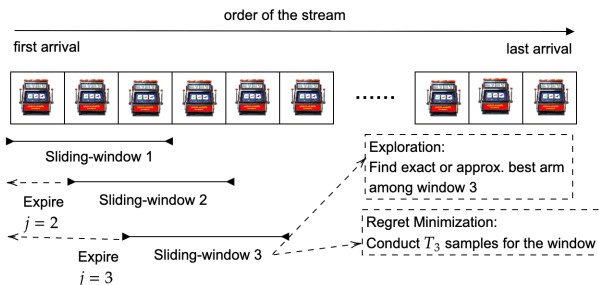

*Figure 1.* An illustration of the sliding-window multi-armed bandits model.

compare with our model are non-stationary bandits (Whittle, 1988) and mortal bandits (Chakrabarti et al., 2008), which also characterize "evolving" bandits over time. However, in those settings, the environment only changes *as a result of arm sampling*, whereas in our model, the environment change is controlled by the data stream.

The separation between data arrival and arm sampling provides a complementary setting to models like non-stationary bandits and mortal bandits. In many applications, e.g., recommending trendy movies, the movies are often on screen for a fixed period of time (*not* as a result of arm pulls). Furthermore, the sliding-window streaming setting takes the *space complexity* into consideration, which is important for modern applications.

## 1.1 Our contributions

We give a comprehensive analysis of pure exploration and regret minimization algorithms for sliding-window streaming MABs. Our results can be summarized in Table 1.

**Pure explorations.** For pure explorations, we studied both *pure exploration*, where the goal is to return the *exact* best arm, and $\varepsilon$-exploration, where the goal is to return an arm whose mean is $\varepsilon$-close to the best. In both notions, the best arm is defined as the arm with the highest mean reward in the *sliding window*. Our main conceptual message is that finding the *exact* best arm is hard without using $\Omega(W)$ arms of memory space, but finding an *approximate* best arm is possible with efficient sample and space.

**Result 1** (Informal statement of Theorems 1 and 2)**.** *The following statements are true for exploration in sliding-window MABs.*

- *Any algorithm that finds the (exact) best arm at any step with probability at least $1/2 + \Omega(1)$ in the sliding-window streaming multi-armed bandits requires $\Omega(W)$ arm memory, even with an unlimited number of arm pulls.*

> - *There exists an algorithm that finds an (approximate) $\varepsilon$-best arm with probability at least $1 - \delta$ at any step with $O(\frac{1}{\varepsilon})$ arm memory and $O(\frac{n}{\varepsilon^2} \log \frac{W}{\delta})$ arm pulls.*

Our results demonstrate a sharp dichotomy between the exact and approximation exploration tasks. For exact exploration, we essentially cannot do anything better than storing the entire sliding window. In contrast, we can identify an approximate best arm with arbitrary constant accuracy using only $O\left(\frac{1}{\varepsilon}\right)$ memory. For a constant choice of $\varepsilon$ (which is usually the case), our algorithm achieves *constant memory* for exploration.

We note that in our algorithms, the number of samples used for each arriving arm is simply $O(\frac{1}{\varepsilon^2} \log \frac{W}{\delta})$, which is independent of $n$. Therefore, the algorithm does not need to know the value of $n$ in advance [3].

**Regret minimization.** For *regret minimization*, a significant challenge is how to *define* regret in the sliding-window model. The most natural definition is to define the regret as the cumulative gap between $\mu^*(t, W)$ and the mean rewards of the pulled arms in each window. Here, $\mu^*(t, W)$ is the mean reward of the optimal arm in the window $W$ at time $t$. However, such a definition has a fatal issue: since the algorithm controls the number of arm pulls before the window moves, the definition of the regret becomes a function of the algorithm, which means it cannot be well-defined.

To address the issue, we introduce the notion of *epoch-wise* regret, which allows for optimal reward sequences that are *independent* of the arm pulls made by the algorithm. Our notion of regret minimization involves dividing the total number of arm pulls, denoted as $T$, into $n - W + 1$ *epochs*. Each epoch $j$ corresponds to a distinct window position (and an arriving arm) and is allocated a prescribed number of arm pulls, represented as $T_j$, constrained within each time window. The sequence $\{T_j\}_{j=1}^{n-W+1}$ is chosen arbitrarily but fixed in advance (*non-adaptive*), and satisfies $\sum_{j=1}^{n-W+1} T_j = T$. Our goal is to minimize the *total regret* across epochs.

We believe that the introduction of the regret notion is a significant contribution; otherwise, there is no obvious way to study regret minimization in sliding-window MABs. Moreover, the epoch-wise regret definition captures many practical scenarios. For instance, in the case of movie recommendations, we treat the "sliding window" as time periods of, e.g., 1-2 months, and we aim to recommend the most relevant movies in each period. Our main conceptual finding for regret minimization is that a memory of $\Omega(W)$ arms is necessary to achieve $o(T)$ regret; furthermore, there is a sharp

---

[3] Except for a certain type of variant in Section 4.3; see the corresponding section for details.

memory-regret transition around the $\Theta(W)$ arm memory.

> **Result 2** (Informal statement of Theorem 3). *Any algorithm that achieves $o(T/W^2)$ regret in the* epoch-wise *regret setting requires $\Omega(W)$ arm memory. Furthermore, there exist algorithms that given a stream of $n$ arms, parameters $T$, $W$, and $\{T_j\}_{j=1}^{n-W+1}$, with $O(W)$ memory achieve $O(\sum_{j=1}^{n-W+1} \sqrt{WT_j})$ regret.*

It is possible to achieve $O(\sqrt{WT})$ regret for specific types of arm pull budgets, which is better than the $O(\sqrt{nT})$ regret in the centralized setting even with unlimited memory. We find the conceptual message quite interesting, and we believe it could serve as an important guideline for related applications. A variant of our regret setting is when the best arm does not expire with the movement of the sliding window. A discussion of this setting can be found in Section D.

**Our techniques.** We start with $\varepsilon$-exploration in sliding-window streaming MABs, in which there are two main technical challenges: the memory constraints and the expiration of arms. An efficient sliding-window algorithm would imply an efficient streaming algorithm (by setting $W = n$); as such, for any MABs technique to work in the sliding-window setting, there must be a streaming algorithm as well. For the majority of MABs techniques, efficient streaming algorithms either do not exist (e.g., elimination-based algorithms (Even-Dar et al., 2006; Karnin et al., 2013)), or it is unclear how to design such algorithms (e.g., non-stationary bandits (Whittle, 1988) and mortal bandits (Chakrabarti et al., 2008)). Therefore, we have to find technical ideas from existing streaming MABs algorithms (e.g. (Assadi & Wang, 2020; Jin et al., 2021; Maiti et al., 2021)).

Most of the algorithms in streaming MABs are either based on amortizing the sample complexity across the stream or using a bucket-based idea to group arms based on the empirical means. The idea of amortization faces a barrier due to the expiration of arms. In particular, for the algorithms that amortize sample complexity in, e.g., (Assadi & Wang, 2020; Jin et al., 2021), the guarantees are only given with respect to the best arm, and the analysis falls apart if the best arm changes due to the sliding window movements. On the other hand, the grouping of empirical means based on multiples of $\varepsilon$ is naturally compatible with the sliding-window model. Here, we can simply discard the expired arms, and the invariant among the buckets helps maintain $\varepsilon$-best arms. This is the main idea for our $\varepsilon$-exploration algorithms.

The main challenge in our lower bound is to find a distribution that forces the algorithm to store virtually all arms. Our idea is to use a distribution of arms with *decreasing mean rewards*. This distribution forces the "useless" arm when the window is at position $t$ to become optimal when the window moves to $t + 1$. Although the distribution is not involved, it

| Task | Space | Sample/Regret | Remark |
|------|-------|---------------|--------|
| Exact Exploration | $\Omega(W)$ | Any | Lower Bound |
| Strong $\varepsilon$-exploration | $O(1/\varepsilon)$ | $\Theta(\frac{n}{\varepsilon^2} \log n)$ | Upper and Lower Bounds |
| Weak $\varepsilon$-exploration | $O(1/\varepsilon)$ | $\Theta(\frac{n}{\varepsilon^2} \log W)$ | Upper Bound |
| Regret minimization | $o(W)$ | $\Omega(T/W^2)$ | Lower Bound |
| | $\Omega(W)$ | $O(\sqrt{W \cdot (n - W) \cdot T})$ | Upper Bound |

*Table 1.* Summary of the results for exploration and regret minimization

crucially uses the sliding-window property to separate from the streaming case, especially given that the latter admits an efficient algorithm with a single-arm memory (Assadi & Wang, 2020). Finally, our regret lower bound follows the same idea, although we need to extend the distribution to more involved ones to ensure the algorithm cannot get "lucky" with certain arms.

**Experiments.** We conducted experiments for both pure exploration and regret minimization applications[4]. For pure exploration, we implemented the $\varepsilon$-best pure exploration algorithm, and for regret minimization, we used the $O(W)$-memory algorithms outlined in Result 2. These are the first algorithms designed to work with multi-armed bandits (MABs) under a sliding-window setting.

In our pure exploration experiments, we tested configurations with $n \in \{1000, 2000, 5000\}$ and $W \in \{10, 20, 50\}$. The results indicate a relatively smooth trade-off between the quality of the returned arm and the memory used. The error exceeded 0.6 in all settings when we employed a memory size of $0.05W$; however, it dropped to below 0.3 with a memory size of $0.3W$. On the other hand, we can easily show that existing algorithms could result in 0.6 error (Section E), and our empirical results essentially mean that with $0.3W$ memory, the error could be reduced by 50%. For the regret minimization experiments, we tested configurations with $n \in \{500, 1000, 2000\}$ and $W \in \{10, 20, 50\}$, while setting the number of pulls for each epoch to $\frac{T}{n - W + 1} = 1000$. The results revealed sharp changes in regret around the memory size $W$, confirming our theoretical predictions. The total regret decreased by more than 50% for most configurations when the memory size increased from $0.05W$ to $W$.

**Additional related work.** Apart from the streaming MABs algorithms, our work is also closely related to settings with evolving and expiring arms, including arm-acquiring bandits (Whittle, 1981), non-stationary bandits (Whittle, 1988),

mortal bandits (Chakrabarti et al., 2008), and the sleeping experts problems (Kleinberg et al., 2010). For instance, in the mortal bandits problem, the arm can expire after a certain number of samples, or it expires with some probability at each step. We remark that although these settings are in a similar spirit to arm expiration, their settings are not directly comparable with ours since we always prioritize the most recent arms in the sliding-window model (i.e., the separation between data arrival and arm pulls). Furthermore, none of these problems considered *memory* constraints, which means their algorithms cannot be directly applied in our setting.

## 2 Problem Definition and Preliminaries

In this section, we give the formal definition of the problems we investigate and some standard technical tools. We start with a formal definition of stochastic MABs.

**Definition 1** (Stochastic multi-armed bandits (MABs) model)**.** In the stochastic multi-armed bandits model, we have a collection of $n$ arms $\{\mathtt{arm}_i\}_{i=1}^n$, and each arm follows a distribution supported on $[0, 1]$ with mean reward $\mu_i$. Each pull of $\mathtt{arm}_i$ returns a sample from the distribution with mean $\mu_i$.

It is important to note that, according to the central limit theorem, sampling from arbitrary distributions supported on $[0, 1]$ is essentially equivalent to sampling from an arbitrary sub-Gaussian distribution (up to a scaling factor). A more in-depth discussion of sub-Gaussian distributions can be found in Section A. We define the sliding-window streaming multi-armed bandits (MABs) as follows.

**Definition 2** (The sliding-window streaming MABs model.)**.** In the sliding-window streaming MABs model, we have a collection of $n$ arms $\{\mathtt{arm}_i\}_{i=1}^n$ arranged in order and a window size $W$. Each arm follows a distribution supported on $[0, 1]$ with mean reward $\mu_i$. The arms arrive one by one in the stream, and we let $\{\mathtt{arm}_i\}_{i=t-W+1}^t$ be the set of valid arms that arrived in the $W$ latest steps. When a new arm arrives, the algorithm can pull the arriving arm and the arms

---

[4]Our code is available on Github: `https://github.com/jhwjhw0123/sliding-window-streaming-MABs`.

in memory. The algorithm can also decide whether to store the new arm in memory or discard it, and the algorithm can discard some arms stored in memory to free up space. Furthermore, any *expired arm*, i.e., the arms outside the sliding window, is *immediately deleted* from the memory. At any point, the collection of arms that the algorithm can access is the arms in memory and the arriving arm.

We emphasize that we deal with the *single-pass* and *worst-case order* scenario, in which the algorithm is only allowed to make *one pass* over the stream, and the arrival order of the arms is specified by an adversary.

**Remark 1.** To keep consistent with the literature in sliding-window streaming algorithms, e.g., (Datar et al., 2002; Datar & Motwani, 2016), we do *not* force the algorithm to discard the expired arms. Nevertheless, our upper and lower bounds in Result 1 and Result 2 do *not* rely on this property. In other words, if we add the condition that the expired arms have to be deleted immediately, the upper and lower bounds in Result 1 and Result 2 still hold. The immediate removal of expired arms from the memory is helpful for applications with private data retention requirements.

We can now define the *sample and space complexity* of a sliding-window streaming MABs algorithm.

**Definition 3** (*Sample complexity*). The *sample complexity* of a sliding-window streaming MABs algorithm is defined as the total number of pulls of the algorithm.

**Definition 4** (*Space complexity*). The *space complexity* of a sliding-window streaming algorithm is defined as the maximum number of arms that we store in the memory at any time during the algorithm.

**Pure exploration.** One of the most natural problems in the MABs problem in the sliding-window model is the *pure exploration* problem, where the algorithm is asked to return the best or near-best arms. In what follows, we discuss the necessary notions before formally defining the pure exploration problems.

**Definition 5** (*Best arm in the window*). Assume that we have a collection of $n$ arms $\{\texttt{arm}_i\}_{i=1}^n$ with means $\mu_i$ and arranged in the streaming arriving order. Let $W$ be the window size and $t$ be the index of the current arriving arm. Then, for any $t \in [n]$, the best arm in the window $\texttt{arm}^*(W, t)$ is the arm with the highest mean $\mu^*(W, t)$ among the $W$ latest arms $\{\texttt{arm}_i\}_{i=t-W+1}^t$.

Note that the notation $\texttt{arm}^*(W, t)$ is a function of $t$ and $W$. We also call the set of arms $\{\texttt{arm}_i\}_{i=t-W+1}^t$ *valid* at time step $t$ for fixed $t$ and $W$.

We are ready to introduce the *pure exploration* problem for the sliding-window streaming MABs model.

**Problem 1** (Exact pure exploration in sliding-window MABs). Given a stream of $n$ arms $\{\texttt{arm}_i\}_{i=1}^n$ and a win-

dow size $W$, we say a sliding-window streaming MABs algorithm ALG solves

- *weak* pure exploration with probability $1 - \delta$ if for any single *fixed* time $t \in [n]$, ALG can output the best arm in the window with probability at least $1 - \delta$.
- *strong* pure exploration with probability $1 - \delta$ if ALG can output the best arm in the window simultaneously at all times $t \in [n]$ with probability $1 - \delta$.

Next, we could analogously define the $\varepsilon$-exploration problem in both the *weak* and the *strong* versions for the sliding-window streaming MABs.

**Problem 2** ($\varepsilon$-exploration in sliding-window MABs). Given a stream of $n$ arms $\{\texttt{arm}_i\}_{i=1}^n$, a window size $W$, and a parameter $\varepsilon$, we say a sliding-window streaming MABs algorithm ALG solves

- *weak* $\varepsilon$-exploration with probability $1 - \delta$ if for any single *fixed* time $t \in [n]$, ALG is able to output an arm with mean reward $\mu$ such that $\mu \geqslant \mu^*(t, W) - \varepsilon$ with probability at least $1 - \delta$.
- *strong* $\varepsilon$-exploration with probability $1 - \delta$ if ALG is able to output an arm with mean reward $\mu$ such that $\mu \geqslant \mu^*(t, W) - \varepsilon$ simultaneously at all times $t \in [n]$ with probability $1 - \delta$.

Here, as defined in Definition 5, $\mu^*(t, W)$ is the mean reward of the best arm in the window.

**Regret minimization.** In Section 1.1, we have discussed the high-level definition for our regret notion in sliding windows, i.e., the epoch-wise regret. We now introduce the formal definition as follow.

**Definition 6** (Regret minimization, epoch-wise regrets). Let $\{\texttt{arm}_i\}_{i=1}^n$ be a set of $n$ arms, and let $W$ and $T$ be the window size and the total number of samples. We divide $T$ into $n - W + 1$ *epochs* with $T_j$ as the number of samples in each epoch, and we have $\sum_{j=1}^{n-W+1} T_j = T$. For the arriving arm with index $j$, the algorithm is required to conduct *exactly* $T_j$ arm pulls among $\{\texttt{arm}_i\}_{i=j-W+1}^j$. We define the regret of the $j$-th epoch as $R^E(j) = \sum_{\tau=1}^{T_j} (\mu^*(W, j) - \mu_{i(\tau)})$, where $i(\tau)$ is the arm index pulled by the algorithm. The total regret is defined as $R_T = \sum_{j=1}^{n-W+1} R^E(j)$, i.e., the regret over the epochs.

## 3  A Lower Bound for Pure Exploration in Sliding-Window MABs

The most natural pure exploration problem is *pure exploration* which asks to return the *best arm*. In the vanilla streaming multi-armed bandits (MABs) model, pure exploration can be solved with $O(n/\Delta_{[2]}^2)$ samples and a single-arm memory, where $\Delta_{[2]}$ represents the difference between the mean of the best and the second-best arms. As such, one would naturally wonder whether the same story applies to

the sliding-window model. In this section, we will show that pure exploration is surprisingly much harder in the sliding-window streams: unless the algorithm uses $\Omega(W)$ space, we cannot obtain any algorithm that solves pure exploration.

The hard instance for our lower bound is a stream with descending mean rewards of arms, i.e., $\mu_1 > \mu_2 > \cdots > \mu_n$ for arms. The optimal solution for the sliding-window MABs would be to select $\texttt{arm}_{n-W+1}$, which is the oldest non-expired arm. However, to always keep the oldest arm that has not expired in the memory, we would naturally need $W$ memory. The following theorem formalizes the above intuitions.

**Theorem 1.** *Any algorithm that given $n$ arms in a sliding-window stream with a window size of $W$, solves the* weak *or* strong *pure exploration problem in sliding-window streaming multi-armed bandits with a probability of at least $1/2 + \Omega(1)$ has a space complexity of at least $\Omega(W)$, even if the sample complexity is unbounded.*

*Proof.* We prove the theorem for weak pure exploration, and since the answer for strong exploration is always valid for weak exploration, the strong exploration problem requires at least the same amount of memory and samples. We also focus on proving the theorem with a success probability of $99/100$ for ease of presentation. By a standard reduction argument, the lower bound can be boosted to work against any algorithms with $1/2 + \Omega(1)$ success probability.

By Yao's minimax principle (Yao, 1977), it is sufficient to prove the lower bound for deterministic algorithms over a hard distribution of inputs. Let $n = 2W$. We construct the instance $\{\texttt{arm}_1\}_{i=1}^n$ such that $\mu_i = 1 - \frac{i}{3W}$. To solve the *weak* pure exploration problem with a probability of at least $\frac{99}{100}$, the algorithm must correctly identify at least $\frac{49}{50}$ of the best arms in the second half of the stream $\{\texttt{arm}_i\}_{i=1}^n$. If the algorithm fails to do this, the overall success probability would drop below $1 \cdot \frac{1}{2} + \frac{49}{50} \cdot \frac{1}{2} = \frac{99}{100}$.

Let $T \subset \{W+1, W+2, \ldots, 2W\}$ represent the collection of times when the algorithm correctly identifies the best arm in the window during the second half of the stream. Define $A = \{\texttt{arm}^*(W, t) | t \in T\}$ as the set of best arms in the window at times $t \in T$. For any $t \in \{W+1, W+2, \ldots, 2W\}$, the best arm in the window $\texttt{arm}^*(W, t)$ should be $\texttt{arm}_{t-W+1}$ because the expected values of the arms monotonically decrease in this instance. Therefore, we have $A = \{\texttt{arm}_{t-W+1} | t \in T\}$. Given that $T \subset \{W+1, W+2, \ldots, 2W\}$ and $|T| \geqslant \frac{49}{50}W$, it follows that $A \subset \{\texttt{arm}_2, \texttt{arm}_3, \ldots, \texttt{arm}_{W+1}\}$ and $|A| = |T| \geqslant \frac{49}{50}W$.

For any $W + 1 \leqslant t < 2W$, $\texttt{arm}^*(W, t) = \texttt{arm}_{t-W+1}$ has already arrived by time $W + 1$. Therefore, for any $t \in T \cap [2W - 1]$, $\texttt{arm}^*(W, t)$ must be stored in memory by time $W+1$ so that it can be returned at time $t$. This means that at least $|A| - 1 = \frac{49}{50}W - 1$ arms must be stored in

memory at time $W + 1$. Hence, according to Yao's minimax principle, the algorithm must have a *space complexity* of at least $\Omega(W)$. $\qquad\square$

## 4  Sliding-Window Algorithms and Lower Bounds for $\varepsilon$-Pure Exploration

Section 3 depicts a very pessimistic picture for the pure exploration of the *best arm* in sliding-window streaming MABs. A natural question to follow is whether we could get positive results using a relaxed notion. A natural candidate for this purpose is the $\varepsilon$-exploration under the $(\varepsilon, \delta)$-PAC framework. Here, instead of returning the *single best* arm, we are allowed to obtain an arm whose gap is within $\varepsilon$ additive to the best, i.e., returning an arm with mean reward $\mu \geqslant \mu^* - \varepsilon$. In this section, we present the bounds for both *strong* and *weak* $\varepsilon$ exploration. Our main results are $i)$. a pure exploration algorithm that solves *weak $\varepsilon$* exploration with probability $1 - \delta$ in the sliding-window streaming MABs model with $O\left(\frac{n}{\varepsilon^2} \log \frac{W}{\delta}\right)$ *sample complexity* and $O\left(\frac{1}{\varepsilon}\right)$ *space complexity*; $ii)$. a lower bound showing that for any algorithm to solve *strong $\varepsilon$* exploration with probability $1/2 + \Omega(1)$ in the sliding-window streaming MABs model, the algorithm has to use $\Omega(\frac{n}{\varepsilon^2} \log \frac{n}{W})$ sample complexity; and $iii)$. a nearly-matching algorithm for *strong $\varepsilon$* exploration with probability $1 - \delta$ in the sliding-window streaming MABs model with $O\left(\frac{n}{\varepsilon^2} \log \frac{n}{\delta}\right)$ *sample complexity* and $O\left(\frac{1}{\varepsilon}\right)$ *space complexity*.

### 4.1  An efficient algorithm for weak $\varepsilon$-pure exploration

We start with introducing a streaming algorithm designed for *weak $\varepsilon$* exploration.

**Theorem 2.** *There exists an algorithm that, given $n$ arms arriving in a sliding-window stream with a window size $W$ and a confidence parameter $\delta$, solves* weak $\varepsilon$ *exploration with a probability of at least $1 - \delta$ using a sample complexity of $O\left(\frac{n}{\varepsilon^2} \log \frac{W}{\delta}\right)$ and a space complexity of $O\left(\frac{1}{\varepsilon}\right)$.*

At a high level, the algorithm follows the idea of partitioning the range $[0, 1]$ into $O\left(\frac{1}{\varepsilon}\right)$ segments ("buckets") of equal length. An arm is considered to belong to a bucket if its mean value falls within the range of that segment. For an arm $\texttt{arm}_i$ that belongs to bucket $B$, any arm $\texttt{arm}'$ that is in a nearby bucket would serve as an $\varepsilon$-approximation of $\texttt{arm}_i$. If we pull each arm an adequate number of times, we can ensure that any arm is placed into a bucket that is close enough to its mean; thus, the non-expired arm from the highest bucket will be an $\varepsilon$-best arm. To optimize memory usage, we store only the latest arm for each bucket instead of all the arms that belong to that bucket. Our algorithm for *weak $\varepsilon$* exploration is presented in Algorithm 1, with number of arm pulls $s = (9/2\varepsilon^2) \cdot \ln 6W/\delta$.

**Algorithm 1** Efficient Algorithm for $\varepsilon$ exploration in Sliding-window Streaming MABs: BUCKET($s$)

---

**Input:** Data stream $\{\text{arm}_i\}_{i=1}^n$, window size $W$, *confidence parameter* $\delta$ and *accuracy parameter* $\varepsilon$

**Input:** Sample complexity: $s = \frac{9}{2\varepsilon^2} \ln \frac{6W}{\delta}$ for weak exploration and $s = \frac{9}{2\varepsilon^2} \ln \frac{6n}{\delta}$ for strong exploration.

**Output:** $\varepsilon$-best arms $\{\widehat{\text{arm}}_i\}_{i=1}^n$.

$N \leftarrow \frac{3}{\varepsilon}$ and initialize $N$ buckets $B_1, B_2, \cdots, B_N$.

**for** *each arriving arm* $\text{arm}_i$ **do**

    Pull $\text{arm}_i$ for $s$ times and evaluate empirical mean $\widehat{\mu}_i$.

    Store $\text{arm}_i$ in $B_j$ such that $(j-1)\frac{\varepsilon}{3} < \widehat{\mu}_i \leqslant j\frac{\varepsilon}{3}$, and discard the arms stored in $B_j$ previously.

    // Expired arms also discarded

    $\widehat{\text{arm}}_i \leftarrow$ the arm stored in $B_k$ such that $k = \max_{i \leqslant N}\{B_i \neq \emptyset\}$.

**end**

**return** $\{\widehat{\text{arm}}_i\}_{i=1}^n$.

---

### 4.2 A lower bound for strong $\varepsilon$-pure exploration

We now discuss the lower bound for *strong $\varepsilon$ exploration* that has an extra $\log n$ factor. In particular, if we show that when $W \ll n$ (e.g., $W = \log n$), there is a lower bound of $\Omega(\frac{n}{\varepsilon^2} \log n)$ samples for strong exploration, it would imply a *separation* between the *weak* and *strong $\varepsilon$ exploration* since the weak exploration only requires $\Omega(\frac{n}{\varepsilon^2} \log W)$ samples by Algorithm 1 BUCKET $\left(\frac{9}{2\varepsilon^2} \ln \frac{6W}{\delta}\right)$. Then we have:

**Lemma 4.1.** *For infinitely many choices of parameters $n$, $\varepsilon$, and $W \leqslant n^{0.99}$, there exists a distribution of arms $\mathcal{D}(n, W, \varepsilon)$ such that any algorithm that solves the* strong *$\varepsilon$ exploration with probability at least $1/2 + \Omega(1)$ on $\mathcal{D}(n, W, \varepsilon)$ requires at least $\Omega(\frac{n}{\varepsilon^2} \log n)$ samples. The lower bound holds even if the algorithm is with unbounded memory.*

The technical statement for Lemma 4.1 is more general and gives $\Omega(\frac{n}{\varepsilon^2} \log \frac{n}{W})$ samples for $W \in [1, n/8]$, although the bound is less informative when $W$ is large. At a high level, our lower bound works by reducing the problem of solving *independent* copies of the $\varepsilon$-best arm identification to the sliding-window streaming $\varepsilon$ exploration case. (Mannor & Tsitsiklis, 2004) proved that $O\left(\frac{n}{\varepsilon^2} \log\left(\frac{1}{\delta}\right)\right)$ pulls are necessary to identify an $\varepsilon$-best arm among $n$ arms with a probability of at least $1 - \delta$. In the slide-window setting, since arms will expire after $W$ time, the information from *disjoint* windows is *independent*. There are $\Theta(\frac{n}{W})$ windows in a sliding-window stream that are disjoint. Since each window requires at least $O\left(\frac{W}{\varepsilon^2} \log\left(\frac{n}{W}\right)\right)$ pulls to solve its exploration with a probability of at least $1 - \Theta\left(\frac{W}{n}\right)$, it follows that $O\left(\frac{n}{\varepsilon^2} \log\left(\frac{n}{W}\right)\right)$ pulls are necessary to achieve *strong $\varepsilon$-exploration* with a probability of at least $1/2 + \Omega(1)$.

### 4.3 An efficient algorithm for strong $\varepsilon$-pure exploration

We introduce a streaming algorithm for *strong $\varepsilon$ exploration*. The algorithm uses essentially the same subroutine as in Algorithm 1, but it uses a larger number of arm pulls $s = \frac{9}{2\varepsilon^2} \ln \frac{6n}{\delta}$ to beat a union bound.

**Lemma 4.2.** *There exists an algorithm that, given $n$ arms arriving in a sliding-window stream with a window size $W$ and a confidence parameter $\delta$, solves* strong *$\varepsilon$ exploration with a probability of at least $1 - \delta$, a sample complexity of $O\left(\frac{n}{\varepsilon^2} \log \frac{n}{\delta}\right)$, and a space complexity of $O\left(\frac{1}{\varepsilon}\right)$.*

The algorithm in Lemma 4.2 is the only algorithm in this paper that requires knowing $n$ in advance. Here, we need to pay an extra $O(\log n)$ factor in the sample complexity of each arm. Such a bound is also necessary, as we prove in Lemma 4.1.

## 5 Regret Minimization in Sliding-Window Streaming MABs

In this section, we investigate *regret minimization* for sliding-window streaming multi-armed bandits (MABs). Recall that in Definition 6, we defined regret minimization with the concepts of *epoch-wise* regret. Here, we have $n - W + 1$ epochs, and we must perform $T_j$ pulls in each epoch. The question is how to minimize the cumulative regret over the entire horizon $[T]$.

The most natural idea is to adapt strategies in streaming MABs, e.g., (Wang, 2023), to get a low regret algorithm. When a new arm arrives, we can use Algorithm 1 with parameter $s = O(\frac{1}{\varepsilon^2} \log n)$. By the guarantees of the streaming algorithm, we will be able to get $\varepsilon$-best arms at any step with high probability. This strategy incurs a regret of $O(\frac{1}{\varepsilon^2} \log n)$ when identifying the $\varepsilon$-best arm during each epoch. Additionally, there is a regret of $O(\varepsilon T_j)$ for the remaining pulls on the $\varepsilon$-best arm we identify within each epoch. As a result, the total regret is $O(\frac{n}{\varepsilon^2} \log n + \varepsilon T)$. The regret is minimized by choosing $\varepsilon = O(\sqrt[3]{\frac{n \log n}{T}})$, which gives a total regret of $O(T^{\frac{2}{3}}(n \log n)^{\frac{1}{3}})$.

Unfortunately, this strategy has a fatal issue: Algorithm 1 requires $O\left(\frac{1}{\varepsilon}\right)$ memory space; and since in most cases $T \gg n \gg W$, the memory of $1/\varepsilon = O(\sqrt[3]{\frac{T}{n \log n}})$ can be much larger than the window size $W$. Thus, it is not immediately clear whether we could get low-regret algorithms with small memory in this setting. It turns out the issue is not just an artifact: we prove a strong lower bound, showing that a total regret of $O\left(\frac{T}{W^2}\right)$ is unavoidable if we only have $o(W)$ space.

**Theorem 3.** *There exists a family of streaming stochastic multi-armed bandit instances such that, for any given pa-*

*rameters $T$, $n$, and $W$, where $T \geqslant n \geqslant 16W$, there exists a collection of budgets $\{T_j\}_{j=1}^{n-W+1}$ such that any single-pass streaming algorithm for a sliding-window stream of length $n$ with a window size $W$ and a memory capacity of $\frac{W-1}{2}$ arms must incur a total expected regret given by $\mathbb{E}[R_T] \geqslant \frac{T}{64W^2}$.*

*Furthermore, there exists an algorithm that given $n$ arms arriving in a stream and parameters $W$ and $\{T_j\}_{j=1}^{n-W+1}$, achieves $O(\sum_{j=1}^{n-W+1} \sqrt{T_j \cdot W})$ total regret with a memory of $W$ arms.*

At a high level, our lower bound is obtained by constructing $W$ arms whose means decrease by a $\frac{1}{W}$ factor followed by $W$ arms with the same mean, and the pattern is repeated over the stream. Since we can only store at most half of these arms, if the best arm in the epoch is missed, the regret for each pull will be at least $\frac{1}{2W}$. This leads to a total regret of $\Omega\left(\frac{T}{W^2}\right)$. Our upper bound is obtained by running UCB-based algorithms on each window. By a simple application of the Cauchy-Schwarz inequality, we can show that the worst-case regret is at most $O(\sqrt{W \cdot (n - W + 1) \cdot T})$.

We also remark that our algorithm does *not* require the full sequence $\{T_j\}_{j=1}^{n-W+1}$ to be known in advance. The algorithm only needs to know $T_j$ when the $j$-th epoch is reached, and the bound $O(\sum_{j=1}^{n-W+1} \sqrt{WT_j})$ holds for any realized sequence satisfying $\sum_{j=1}^{n-W+1} T_j = T$. This represents a certain level of "adaptivity" of our algorithm.

## 6 Experiments

We conduct experiments focusing on both $\varepsilon$-exploration and regret minimization in the sliding-window streaming setting. Our primary empirical finding is that, consistent with our theoretical results, both the $\varepsilon$-exploration and regret minimization algorithms exhibit trade-offs between memory usage and quality/regret. Furthermore, our algorithms significantly outperform baseline algorithms using (simple adaptations of) streaming MABs algorithms.

We briefly discuss the experiments related to the $\varepsilon$-exploration and regret minimization algorithms in epoch-wise settings. Additional experimental results can be found in Section F.

**The data.** We use both **synthetic data** and **real-world data** with streams of arms to conduct our experiments. Due to space limits, we focus on the experiments with synthetic data in this section, and defer the results with real-world datasets to Section F. For the synthetic data, we use different types of instances for exploration and regret minimization as follows.

- For exploration, we sample $n$ arms with the distribution

$\mathsf{Bern}(p)$ such that $p$ is from a uniform distribution[5]. We note that the "uniform" types of instances are more suitable for $\varepsilon$-exploration since the quality decrement of the returned arms could be better captured. We use $n \in \{1000, 2000, 5000, 10000\}$ and $W \in \{20, 50, 100, 200\}$ for $\varepsilon$-exploration experiments.

- For regret minimization, we need instance distributions *consistent* with our instance distribution in Section 5. For the epoch-wise regret minimization, we sample $n - n/W$ arms with distribution $\mathsf{Bern}(0.25)$ and $n/W$ arms with distribution $\mathsf{Bern}(0.95)$. We then permute the arms uniformly. Due to constraints on running time, we use $n \in \{500, 1000\}$ and $W \in \{20, 50\}$ for regret minimization experiments.

**The algorithms.** We conduct experiments for both exploration and regret minimization. To mitigate the noise from randomness, for each parameter setting, we conduct 10 **independent runs of experiments and take the average**, and we report the error ranges.

We first implement and test our $\varepsilon$-exploration algorithm (Algorithm 1). For our baseline, we use an adaptation of the streaming algorithm as follows: we utilize a top-$k$ algorithm that employs $k$ memory slots to store the $k$ arms with the highest rewards up to the current timestamp. The sampling strategy is to simply sample each arm $O(n \log n/\varepsilon^2)$ times, and return the top $k$ arms with the highest empirical rewards. We further remove any arm from memory if it expires, in accordance with our sliding-window setting. The baseline algorithm returns the unexpired arm with the highest empirical reward from its memory; if there are no valid arms in memory, it does not return anything.

To implement the regret minimization algorithm with arbitrary memory size (which might be smaller than $W$), we adapt the algorithm in Section 5 in the following manner. In cases where the number of arms $m$ is less than the $W$-arm memory capacity, we implement reservoir sampling as follows. Once the memory is full, for each new arm that arrives, we flip a coin with a bias of $m/t$ (where $t$ is the index of the arriving arm) to determine whether to include the arm in memory. If we decide to admit the new arm, we randomly discard one of the existing arms currently in memory. For our baseline, we utilize the same algorithm we employed for exploration, which retains the top-$k$ arms up to the current timestamp. We then exploit the remaining pulls by selecting the valid arm that has the highest empirical reward. The value of $\varepsilon$ of the baseline algorithm is calculated from the available memory, i.e., we have a memory of $1/\varepsilon$ arms.

The baseline algorithms used in our experiments are adaptations of existing streaming algorithms. In particular, both the exploration and regret-minimization baselines rely on

---

[5]We use $\mathsf{Bern}(p)$ to denote Bernoulli distribution with mean $p$.

the same streaming exploration principle of maintaining a small set of arms with good empirical rewards. However, algorithms designed for the streaming setting cannot be directly used in the sliding-window model due to the expiration of arms. As such, our adaptations are the "natural heuristics" based on existing streaming algorithms. Our experiments show that, even with these adaptations, the performances of the streaming algorithms still demonstrate a large gap compared to the algorithms designed specifically for the sliding-window setting.

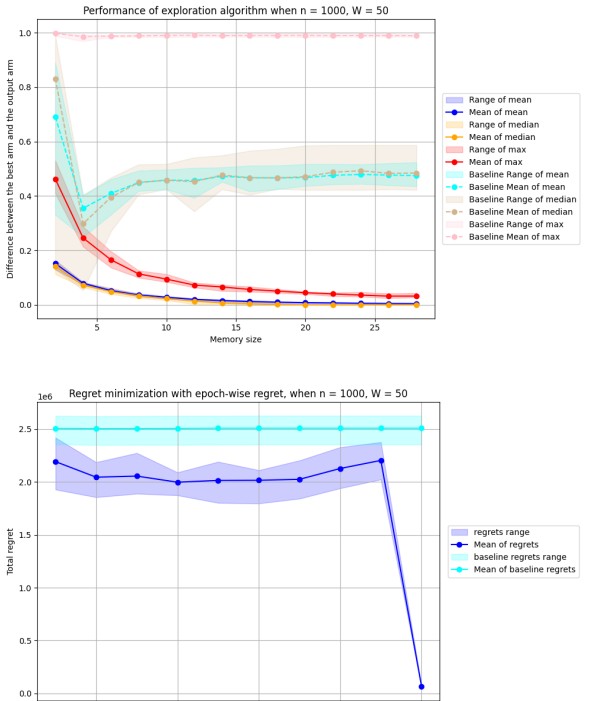

*Figure 2.* The performances of $\varepsilon$-exploration and regret minimization for synthetic data, $n = 1000$, $W = 50$.

**Summary of the experiments.** Due to space limits, we show and discuss the empirical results for $\varepsilon$-exploration and regret minimization on synthetic data for the $n = 1000$ and $W = 50$ case. Additional results are given in Section F.

Figure 2 illustrates the max, median, and mean differences between the reward of the arm returned by the algorithm and the actual best arm across all timestamps (top) and the memory-regret trade-offs of the algorithms (bottom). The figures clearly demonstrate a significant gap between the baseline algorithms and our algorithms. In particular, for both problems, increasing the memory of the baseline algorithms does *not* bring significant benefit for the performances. This gap highlights the crucial distinction between vanilla streaming multi-armed bandits (MABs) and sliding-window MABs. It suggests that the concept of expiration is

vital in the sliding-window model, necessitating the development of specialized sliding-window algorithms to achieve optimal performance.

## 7 Conclusion and Future Work

In this work, we provide a comprehensive study of multi-armed bandits (MABs) in the sliding-window model. Our results establish the fundamental hardness of online learning in the sliding-window MABs model, and we provide important insights for exploration and regret minimization, which can be extremely useful in practice. For instance, by our conceptual message, for exploration tasks in the sliding-window model, we should use $\varepsilon$-exploration rather than finding the *exact* best arm. There are several open directions to follow up on our work. For instance, one appealing question is the *multi-pass* setting: if the algorithm is allowed to make multiple passes over the stream, it might be possible for the algorithm to achieve better memory efficiency. Another appealing question is to ask about gap-dependent regret bounds in the sliding-window streaming model. While gap-dependent bounds are well understood in centralized MABs (Auer et al., 2002), the streaming case is substantially more challenging due to the memory constraint, and a recent work of Ye et al. (2025) is the first to obtain gap-dependent regret bounds in the streaming setting. As such, exploring whether similar bounds in Ye et al. (2025) can be applied to the sliding-window setting is an intriguing direction. Finally, the sliding-window model for other variants of MABs, e.g., the linear bandits and combinatorial bandits, can be additional interesting directions to pursue.

## Acknowledgments

The authors thank the anonymous ICML reviewers for helpful comments. Part of the work was done when Chen Wang was at Rice University and Texas A&M University. Vladimir Braverman is supported in part by NSF CNS-2528780. Samson Zhou is supported in part by NSF CCF-2335411. Samson Zhou gratefully acknowledges funding provided by the Oak Ridge Associated Universities (ORAU) Ralph E. Powe Junior Faculty Enhancement Award.

## Impact Statement

This paper studied the theoretical foundations of online learning. While there are many potential downstream societal consequences of our work, none of them is *immediate* due to the theoretical nature of the work. Therefore, we do *not* believe any of the societal aspects must be specifically highlighted here.

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

# A    Technical Preliminaries

In this section, we provide a number of technical preliminaries for our paper.

**Concentration inequalities.**    We use some standard concentration inequalities in the proof of our results. We provide these inequalities for completeness.

**Proposition A.1** (Chernoff-Hoeffding bound)**.** *Let* $X_1, X_2, \cdots, X_m$ *be a sequence of independent discrete random variables bounded in the range* $[0, 1]$. *Define* $S_m = \sum_{i=1}^{m} X_i$, *then*

$$\mathbf{Pr}\left[|S_m - \mathbb{E}\left[S_m\right]| \geqslant t\right] \leqslant 2 \cdot \exp\left(-\frac{2t^2}{m}\right).$$

We also use the following direct corollaries of the Chernoff-Hoeffding bound.

**Proposition A.2.** *Let* `arm` *be an arm with mean* $\mu$. *We pull the arm* $\frac{K}{\theta^2}$ *times to obtain empirical mean* $\widehat{\mu}$. *Then,*

$$\mathbf{Pr}\left[|\mu - \widehat{\mu}| \geqslant \theta\right] \leqslant 2 \cdot \exp\left(-2K\right).$$

**Proposition A.3.** *Let* `arm`$_1$ *and* `arm`$_2$ *be two different arms with means* $\mu_1$ *and* $\mu_2$. *Suppose* $\mu_1 - \mu_2 \geqslant \theta > 0$ *and we pull each arm* $\frac{K}{\theta^2}$ *times to obtain empirical rewards* $\widehat{\mu}_1$ *and* $\widehat{\mu}_2$. *Then,*

$$\mathbf{Pr}\left[\widehat{\mu}_1 \leqslant \widehat{\mu}_2\right] \leqslant 2 \cdot \exp\left(-\frac{K}{4}\right).$$

We emphasize that while we assumed the support of each arm is within the range $[0, 1]$, all the algorithmic results in this paper also apply if the arms follow a $\sigma^2$-sub-Gaussian distribution for $\sigma = O(1)$. This is possible because the probability guarantee for the case where the arms have support in $[0, 1]$ is derived from the Chernoff-Hoeffding bound (Proposition A.1). Although a sub-Gaussian distribution does not satisfy the conditions for the Chernoff-Hoeffding bound, it still satisfies a similar inequality that can provide a probability guarantee as well. We chose to define the support of all arms as $[0, 1]$ to align with previous research and for the sake of simplicity.

# B    The Complete Details for Results in Section 4

In this section, we provide the complete details (missing algorithms and analysis) we discussed in Section 4.

## B.1    The analysis of Theorem 2 and Algorithm 1

We now proceed to the analysis of Algorithm 1. The following lemma establishes the *space complexity* of the algorithm.

**Lemma B.1.** *The space complexity of* BUCKET $\left(\frac{9}{2\varepsilon^2} \ln \frac{6W}{\delta}\right)$ *is* $O\left(\frac{1}{\varepsilon}\right)$.

*Proof.* Since we discard the previous arm when storing a new arm in a bucket, each bucket will contain at most one arm during the execution of the algorithm. Therefore, the space complexity of the algorithm is bounded by $N = \frac{3}{\varepsilon}$, which is $O\left(\frac{1}{\varepsilon}\right)$. $\qquad\square$

The following lemma provides a bound on the sample complexity of the algorithm.

**Lemma B.2.** *The sample complexity of* BUCKET $\left(\frac{9}{2\varepsilon^2} \ln \frac{6W}{\delta}\right)$ *is* $O\left(\frac{n}{\varepsilon^2} \log \frac{W}{\delta}\right)$.

*Proof.* Since we sample each arm $l = \frac{9}{2\varepsilon^2} \ln \frac{6W}{\delta}$ times within the algorithm, the *sample complexity* is given by $n \cdot l = O\left(\frac{n}{\varepsilon^2} \log \frac{W}{\delta}\right)$. $\qquad\square$

Finally, we prove the correctness of the algorithm.

**Lemma B.3.** *At any time* $t \in [n]$, *the arm* $\widehat{arm}_t$ *outputted by the algorithm* BUCKET $\left(\frac{9}{2\varepsilon^2} \ln \frac{6W}{\delta}\right)$ *is an* $\varepsilon$-*best arm for* `arm`$^*(t, W)$ *with a probability of at least* $1 - \delta$.

*Proof.* At any time $t \in [n]$, let $\text{arm}^*(t, W) = \text{arm}_k$ and $\widehat{\text{arm}}_t = \text{arm}_m$. Additionally, let $b_i$ be the index of the correct bucket to which $\text{arm}_i$ belongs; that is, $\mu_i \in (b_i - 1)\frac{\varepsilon}{3}, b_i \frac{\varepsilon}{3}]$. We also define $b_i'$ as the index of the bucket where $\text{arm}_i$ is stored upon arrival; thus, $\widehat{\mu}_i \in ((b_i' - 1)\frac{\varepsilon}{3}, b_i' \frac{\varepsilon}{3}]$.

Let $\mathcal{E}_i$ be the event that $|b_i - b_i'| \leqslant 1$, indicating that $\text{arm}_i$ is stored in a bucket close to its correct bucket. By Proposition A.2, we have

$$\begin{aligned} \mathbf{Pr}\left[\neg \mathcal{E}_i\right] = \mathbf{Pr}\left[|b_i - b_i'| > 1\right] \\ \leqslant \mathbf{Pr}\left[|\widehat{\mu}_i - \mu_i| > \frac{\varepsilon}{3}\right] \\ \leqslant 2 \cdot \exp\left(-2l \cdot \frac{\varepsilon^2}{9}\right) = \frac{\delta}{3W}. \end{aligned}$$

The output $\widehat{\text{arm}}_t$ must be an $\varepsilon$-best arm if the following events occur at time $t$:

- $\mathcal{F}_1$: There exists some arm stored in $B_{b_k-1} \cup B_{b_k} \cup B_{b_k+1}$;

- $\mathcal{F}_2$: All the buckets $B_j$ for $j > b_k + 1$ are empty;

- $\mathcal{F}_3$: $|b_m - b_m'| \leqslant 1$.

We will output an arm from $\{B_{b_k-1}, B_{b_k}, B_{b_k+1}\}$ if both $\mathcal{F}_1$ and $\mathcal{F}_2$ hold. $\mathcal{F}_3$ guarantees that the output $\widehat{\text{arm}}_t$ is stored in a bucket close to its correct bucket. If all three events occur simultaneously, we have $b_m' \in \{b_k - 1, b_k, b_k + 1\}$ and $|b_m' - b_m| \leqslant 1$. Therefore, $|b_m - b_k| \leqslant 2$, leading to $|\mu_k - \mu_m| \leqslant 3 \cdot \frac{\varepsilon}{3} = \varepsilon$. This means that $\widehat{\text{arm}}_t$ is an $\varepsilon$-approximation of $\text{arm}^*(t, W)$.

We can analyze the probabilities of each event:

$\mathbf{Pr}\left[\mathcal{F}_1\right] \geqslant \mathbf{Pr}\left[\mathcal{E}_k\right] \geqslant 1 - \frac{\delta}{3W}$. This is because if $\mathcal{E}_k$ occurs, we will store $\text{arm}^*(t, W) = \text{arm}_k$ in bucket $B_{b_k'}$, where $|b_k' - b_k| \leqslant 1$. Since $\text{arm}^*(t, W)$ is the best arm at time $t$, it cannot expire, implying we will not drop the arm stored in $B_{b_k'}$ due to expiration. Thus, $B_{b_k'}$ remains non-empty.

$\mathbf{Pr}\left[\mathcal{F}_2\right] \geqslant \mathbf{Pr}\left[\cup_{j=t-W+1}^t \mathcal{E}_t\right] \geqslant 1 - W \cdot \frac{\delta}{3W}$. If all $\mathcal{E}_j, j \in [t - W + 1, t]$ occur, each arm will be stored in a bucket near its correct bucket. Since $\text{arm}^*(t, W) = \text{arm}_k$ is the best arm at time $t$, we have $b_j \leqslant b_k$ for any $j \in [t - W + 1, t]$. Therefore, $b_j' \leqslant b_k + 1$ for any $j$ when $\mathcal{E}_j$ occurs, thus ensuring all buckets $B_j$ for $j > b_k + 1$ are empty.

$\mathcal{F}_3$ is simply the same event as $\mathcal{E}_m$, so $\mathbf{Pr}\left[\mathcal{F}_3\right] \geqslant 1 - \frac{\delta}{3W}$.

Consequently, we obtain:

$$\mathbf{Pr}\left[\mathcal{F}_1 \cap \mathcal{F}_2 \cap \mathcal{F}_3\right] \geqslant 1 - (2 + W) \cdot \frac{\delta}{3W} \geqslant 1 - \delta.$$

$\square$

## B.2 The analysis of Lemma 4.1

We now prove Lemma 4.1 with the general $\Omega(\frac{n}{\varepsilon^2} \cdot \log \frac{n}{W})$ sample lower bound (this directly implies the lower bound when $W \leqslant n^{0.99}$). For ease of presentation, we work with $99/100$ success probability for the lower bound, and a standard argument can generalize the lower bound to work against any algorithm with $1/2 + \Omega(1)$ success probability.

We employ the *sample complexity* lower bound established by (Mannor & Tsitsiklis, 2004) to prove our lemma. We provide the proposition for completeness.

**Proposition B.1** ((Mannor & Tsitsiklis, 2004))**.** *There exist positive constants $c_1, c_2, \varepsilon_0$ and $\delta_0$, such that for every $n \geqslant 2$, $\varepsilon \in (0, \varepsilon_0)$ and $\delta \in (0, \delta_0)$, and for every algorithm outputs $\varepsilon$-best arm with probability at least $1 - \delta$, there exists some $\mu = (\mu_1, \mu_2, \cdots, \mu_n) \in [0, 1]^n$ such that*

$$\mathbb{E}_\mu[T] \geqslant c_1 \frac{n}{\varepsilon^2} \log\left(\frac{c_2}{\delta}\right).$$

*$T$ is the number of pulls used in the algorithm. $\mathbb{E}_\mu[T]$ is the expectation of $T$ when arms have means $\mu = (\mu_1, \mu_2, \cdots, \mu_n)$.*

*In particular, $\varepsilon_0$ and $\delta_0$ can be taken equal to $1/8$ and $\frac{e^{-4}}{4}$, respectively.*

We assert that any algorithm capable of solving the *strong $\varepsilon$-exploration* for a stream of $n$ arms, with a window size $W$, and achieving a success probability of at least $99/100$, can be modified to create an algorithm that addresses $\Theta\left(\frac{n}{W}\right)$ independent $\varepsilon$-exploration of $W$ arms concurrently, also with a success probability of at least $99/100$. Moreover, the *sample complexity* of the latter algorithm will be less than or equal to the *sample complexity* of the former algorithm.

Specifically, consider sets $X_i$, for $i \in \left[\frac{n}{2W}\right]$. There are $\frac{n}{2W}$ such sets, where each set $X_i$ consists of $W$ arms. Let $Z$ denote a set that contains $W$ arms, each of which consistently returns 0. We can construct a stream $S = (Z, X_1, Z, X_2, \cdots, Z, X_{\frac{n}{2W}})$. By employing an algorithm designed to solve the *strong $\varepsilon$-exploration* task on the stream $S$, we can simultaneously solve the $\varepsilon$-exploration problem for all sets $X_i$.

**Lemma B.4.** *If an algorithm* ALG *exists that successfully solves the* strong $\varepsilon$-exploration problem for a stream of $n$ arms *with a window size $W$ with a probability of at least $99/100$, and has a sample complexity $m$, then there exists another algorithm* ALG$'$ *that can solve $\frac{n}{2W}$ independent $\varepsilon$-exploration problems for $W$ arms simultaneously. This new algorithm* ALG$'$ *will have a sample complexity $m'$ such that $m' \leqslant m$, while also achieving a success probability of at least $99/100$.*

*Proof.* We demonstrate the lemma by providing a framework, Algorithm 2, which generates the algorithm ALG$'$ based on the algorithm ALG.

---

**Algorithm 2** ALG$'$: Algorithm Transformation

---

**Input:** Arms $\{\text{arm}_{i,j}\}_{i \in [\frac{n}{2W}], j \in [W]}$, algorithm ALG
**Output:** A set of arms $\{\widehat{\text{arm}}_i\}_{i \in [\frac{n}{2W}]}$, where $\widehat{\text{arm}}_i$ is an $\varepsilon$-best arm of $\{\widehat{\text{arm}}_{i,j}\}_{j \in [W]}$
Let $\text{arm}_0$ be the arm always return 0 **for** $i \leftarrow 1$ *to* $\frac{n}{2W}$ **do**
    **for** $j \leftarrow 1$ *to* $W$ **do**
        $\text{arm}'_{(i-1)\cdot 2W + j} \leftarrow \text{arm}_0$
        $\text{arm}'_{i \cdot 2W + j} \leftarrow \text{arm}_{i,j}$
    **end**
**end**
Build stream $S = \{\text{arm}'_k\}_{k=1}^n$   $\{\widetilde{\text{arm}}_k\}_{k=1}^n \leftarrow$ ALG$(S)$ **for** $i \leftarrow 1$ *to* $\frac{n}{2W}$ **do**
    $\widehat{\text{arm}}_i \leftarrow \widetilde{\text{arm}}_{i \cdot 2W}$
**end**
**return** $\{\widehat{\text{arm}}_i\}_{i \in [\frac{n}{2W}]}$

---

By the construction of $S$, an $\varepsilon$-best arm at time $i \cdot 2W$ must also be an $\varepsilon$-best arm among the set $\{\widehat{\text{arm}}_{i,j}\}_{j \in [W]}$. Consequently, if ALG successfully solves the *strong $\varepsilon$-exploration* problem on $S$, then the set $\{\widehat{\text{arm}}_i\}_{i \in [\frac{n}{2W}]}$ must be the $\varepsilon$-best arm for the arms $\{\text{arm}_{i,j}\}_{i \in [\frac{n}{2W}], j \in [W]}$. Since ALG accomplishes the *strong $\varepsilon$-exploration* task on $S$ with a probability of at least $99/100$, it follows that ALG$'$ solves the pure exploration problem on the arms $\{\text{arm}_{i,j}\}_{i \in [\frac{n}{2W}], j \in [W]}$ with the same probability.

Furthermore, since $\text{arm}_0$ is merely a virtual arm, the actual number of pulls by ALG$'$ is equivalent to the pulls used on the real arms $\{\text{arm}_{i,j}\}_{i \in [\frac{n}{2W}], j \in [W]}$. Therefore, the number of pulls made by ALG$'$ is at most equal to the number of pulls made by ALG when solving the *strong $\varepsilon$-exploration* problem on $S$. Hence, we can conclude that $m' \leqslant m$. $\qquad\square$

The following is a technical lemma that states a "direct sum" type of bound for solving $k$ independent copies of the same problem.

**Lemma B.5.** *Let $f$ be a function to compute, and let $\mathcal{H}$ be a distribution from which the inputs of $f$ are sampled. Suppose that solving $f$ over the distribution $\mathcal{H}$ with probability $1 - \delta$ takes $\Omega(q \cdot \log(\frac{1}{\delta}))$ queries on the input. Furthermore, let $\widetilde{\mathcal{H}} = (\mathcal{H}_1, \mathcal{H}_2, \cdots, \mathcal{H}_k)$ be a distribution over $k$ independent copies of $\mathcal{H}$. Then, any algorithm* ALG *that computes $f$ on all copies with probability at least $99/100$ has to make $\Omega(k \cdot q \cdot \log k)$ total queries.*

*Proof.* The lemma follows from a direct calculation of the success probability, and we provide the proof for the purpose of completeness. Define $\mathcal{E}_i, i \in [k]$ as the event that ALG successfully computes $f$ on the $i$-th copy of $\mathcal{H}$, and define $\mathcal{E}$ as the

event that *all* copies of $f$ are correctly computed. We have that

$$
\begin{aligned}
\Pr\left(\mathcal{E}\right) &= \Pr\left(\cap_{i=1}^{k}\mathcal{E}_i\right) \\
&= \prod_{i=1}^{k}\Pr\left(\mathcal{E}_i \mid \cap_{j=1}^{i-1}\mathcal{E}_j\right) && \text{(by the law of total probability)} \\
&= \prod_{i=1}^{k}\Pr\left(\mathcal{E}_i\right). && \text{(by the independence)}
\end{aligned}
$$

Therefore, by using the condition that success probability is at least $99/100$, we have that

$$
\sum_{i=1}^{k}\log\left(\Pr\left(\mathcal{E}_i\right)\right) = \log\left(\Pr\left(\mathcal{E}\right)\right) \geqslant \sigma - 1
$$

for some $\sigma \in (0.9, 1)$. We claim that for at least $k/100$ indices of $i \in [k]$, there must be $\log(\Pr(\mathcal{E}_i)) \geqslant \log(1 - \frac{\sigma}{k})$. Otherwise, the total success probability is at most

$$
\begin{aligned}
\frac{99k}{100}\cdot\log(1-\frac{\sigma}{k}) + \frac{k}{100}\cdot\log(1) &= \frac{99k}{100}\cdot\left(\frac{\ln(1-\frac{\sigma}{k})}{\ln 2}\right) \\
&\leqslant \frac{99k}{100\ln 2}\cdot\left(-\frac{\sigma}{k}\right) && \text{(using } \ln(1+x)\leqslant x) \\
&= -\frac{99\sigma}{100\ln 2} < -1.28 < \sigma - 1. && \text{(by } \sigma > 0.9)
\end{aligned}
$$

Since solving each $f$ with probability $1-\delta$ requires $\Omega(q\cdot\log(\frac{1}{\delta}))$ queries, solving $k/100$ indices with probability at least $1-\sigma/k$ requires

$$
\frac{k}{100}\cdot q\cdot\log(\frac{k}{\sigma}) = \Omega(k\cdot q\cdot\log k)
$$

queries, which is as desired. $\qquad\square$

***Finalizing the proof of Lemma 4.1.*** Consider any algorithm ALG that successfully solves the *strong $\varepsilon$-exploration* problem with a probability of at least $99/100$ on $\mathcal{D}(n, W, \varepsilon)$. Let $T_{\mathrm{ALG}}$ represent the number of pulls executed by this algorithm. We will define ALG$'$ as the algorithm derived from ALG using Algorithm 2, and let $T_{\mathrm{ALG}'}$ be the corresponding number of pulls for ALG$'$.

According to Lemma B.4, ALG$'$ is capable of solving $\frac{n}{2W}$ independent $\varepsilon$-exploration tasks involving $W$ arms simultaneously, and it holds that $\mathbb{E}\left[T_{\mathrm{ALG}'}\right] \leqslant \mathbb{E}\left[T_{\mathrm{ALG}}\right]$.

Since it is necessary to make $\Omega\left(\frac{W}{\varepsilon^2}\cdot\log\left(\frac{1}{\delta}\right)\right)$ pulls to solve the $\varepsilon$-exploration of $W$ arms with a probability of at least $1-\delta$, we can employ Lemma B.5 to conclude that $\Omega\left(\frac{n}{\varepsilon^2}\log\frac{n}{W}\right)$ samples are required for the algorithm ALG. $\qquad\square$

## B.3 The analysis for Lemma 4.2

The algorithm subroutines still follow Algorithm 1, and we employ more arm pulls $l = \frac{9}{2\varepsilon^2}\ln\frac{6n}{\delta}$ compared to the weak exploration. This adjustment allows us to effectively apply a union bound across $n$ arms. The increased pulling size not only ensures that we can accurately identify all the $\epsilon$-best arms simultaneously with high probability, but it also results in higher sample complexity.

The following claim establishes bounds on both the *space complexity* and *sample complexity* of this algorithm.

**Lemma B.6.** *The space complexity of the* BUCKET $\left(\frac{9}{2\varepsilon^2}\ln\frac{6n}{\delta}\right)$ *is* $O\left(\frac{1}{\varepsilon}\right)$, *and the sample complexity is* $O\left(\frac{n}{\varepsilon^2}\log\frac{n}{\delta}\right)$.

The proof follows the same reasoning as the proofs of Lemma B.1 and Lemma B.2, and we skip the details to avoid repetitions. Next, we will demonstrate the correctness of the algorithm. We use the notation $\mu(\texttt{arm})$ to denote the mean of the arm.

**Lemma B.7.** *Let $A = \{\widehat{arm}_t\}_{t=1}^n$ be the set of arms outputted by the algorithm, and let $A' = \{arm^*(W,t)\}_{t=1}^n$ represent the set of best arms. Then, with a probability of at least $1 - \delta$, it holds that $\mu(\widehat{arm}_t) \geqslant \mu(arm^*(W,t)) - \varepsilon$ for all time $t \in [n]$.*

*Proof.* Let $b_i$ denote the index of the correct bucket to which arm $arm_i$ belongs. In other words, we have $\mu_i \in \left((b_i - 1)\frac{\varepsilon}{3}, b_i\frac{\varepsilon}{3}\right]$. We define $b_i'$ as the index of the bucket where $arm_i$ is stored upon its arrival, which implies $\widehat{\mu}_i \in \left((b_i' - 1)\frac{\varepsilon}{3}, b_i'\frac{\varepsilon}{3}\right]$.

Let $\mathcal{E}_i$ be the event that $|b_i - b_i'| \leqslant 1$, indicating that $arm_i$ is stored in a bucket close to its correct bucket. According to Proposition A.2, we have:

$$\mathbf{Pr}\left[\neg \mathcal{E}_i\right] = \mathbf{Pr}\left[|b_i - b_i'| > 1\right] \leqslant \mathbf{Pr}\left[|\widehat{\mu}_i - \mu_i| > \frac{\varepsilon}{3}\right] \leqslant 2 \cdot \exp\left(-2l \cdot \frac{\varepsilon^2}{9}\right) = \frac{\delta}{3n}.$$

By applying the union bound, we can express this as:

$$\mathbf{Pr}\left[\cap_{i=1}^n \mathcal{E}_i\right] = 1 - \mathbf{Pr}\left[\neg \cap_{i=1}^n \mathcal{E}_i\right] = 1 - \mathbf{Pr}\left[\cup_{i=1}^n \neg \mathcal{E}_i\right] \qquad \text{(By De Morgan's Law)}$$

$$\geqslant 1 - \sum_{i=1}^n \mathbf{Pr}\left[\neg \mathcal{E}_i\right] \geqslant 1 - \sum_{i=1}^n \frac{\delta}{3n} = 1 - \frac{\delta}{3}.$$

If the event $\cap_{i=1}^n \mathcal{E}_i$ occurs, it means each arm is placed in a bucket $b_i'$ that is close to its correct bucket $b_i$, satisfying $|b_i - b_i'| \leqslant 1$.

For any $t \in [n]$, suppose that $\widehat{arm}_t = arm_{i_t}$ and $arm^*(W,t) = arm_{j_t}$. Therefore, we have $|b_{i_t} - b_{i_t}'| \leqslant 1$ and $|b_{j_t} - b_{j_t}'| \leqslant 1$. Additionally, since $arm^*(W,t) = arm_{j_t}$ does not expire at time $t$, it follows that the bucket $B_{j_t} \neq \emptyset$ at time $t$. Given that the arm returned by the algorithm at time $t$ is $\widehat{arm}_t = arm_{i_t}$, it must be that $b_{i_t}' \geqslant b_{j_t}'$. Thus, we can derive:

$$b_{i_t} \geqslant b_{i_t}' - 1 \geqslant b_{j_t}' - 1 \geqslant b_{j_t} - 2.$$

Consequently, we obtain $\mu(\widehat{arm}_t) \geqslant \mu(arm^*(W,t)) - \frac{2}{3}\varepsilon$. □

## C The Complete Details for Results in Section 5

We provide the proof of Theorem 3 in this section.

*Proof.* We proceed with the lower bound proof first. According to Yao's minimax principle (Yao, 1977), it is sufficient to establish the lower bound for deterministic algorithms under a challenging distribution of inputs. Consider the following distribution of $n$ arms.

---

**EPOCH$(n, W)$: A hard distribution with $n$ arms for epoch-wise regret minimization**

1. For $i = k \cdot 2W + j$, where $j \in [W]$, $\mu_i = 1 - \frac{j}{W}$.

2. With probability $\frac{1}{W}$, choose $h \in [W]$ uniformly. For $i = k \cdot 2W + W + j$, where $j \in [W]$, $\mu_i = 1 - \frac{2h+1}{2W}$.

---

Since we have only $\frac{W-1}{2}$ space available, there exists an arm $arm_i$ where $i \in [k \cdot 2W + 1, k \cdot 2W + W]$ that we cannot store at time $k \cdot 2W + W$. Let $\mathcal{E}_i$ be the event where the $W$ subsequent arms all have the same mean of $1 - \frac{2h+1}{2W}$, with $h \equiv i$ (mod $2W$), but $arm_i$ is not stored at time $k \cdot 2W + W$. When event $\mathcal{E}_k$ occurs, $arm_i$ is the best arm at time $i + W - 1$. As we missed $arm_i$, this will induce at least $\frac{1}{2W} \cdot \frac{T}{n-W+1}$ regret during this epoch.

Let

$$\mathcal{F}_k = \bigcup_{i=k \cdot 2W+1}^{k \cdot 2W+W} \mathcal{E}_i.$$

The event $\mathcal{F}_k$ represents that at least one of the events $\mathcal{E}_i$ occurs between the times $[k \cdot 2W + 1, (k+1) \cdot 2W]$. Since at least half of the arms among $\{\text{arm}_{k \cdot 2W+1}, \cdots, \text{arm}_{k \cdot 2W+W}\}$ are not stored at time $k \cdot 2W + W$, we have $\mathbf{Pr}\left[\mathcal{F}_k\right] \geqslant \frac{1}{2}$.

Let $X_k$ be the random variable indicating whether $\mathcal{F}_k$ occurs. The total number of such events that occur is $Y = \sum_{k=1}^{m} X_k$, where $m = \lfloor \frac{n}{2W} \rfloor$. Since $X_i$ are independent Bernoulli random variables with probability $p \geqslant \frac{1}{2}$, $Y$ follows a binomial distribution $Y \sim \text{Bino}(m, p)$. Let $Z \sim \text{Bino}(m, \frac{1}{2})$.

We can analyze the probability as follows:

$$\mathbf{Pr}\left[Y \leqslant \frac{m}{4}\right] \leqslant \mathbf{Pr}\left[Z \leqslant \frac{m}{4}\right] \qquad \text{(since } p \geqslant \frac{1}{2}\text{)}$$

$$\leqslant \mathbf{Pr}\left[\left|Z - \frac{m}{2}\right| \geqslant \frac{m}{4}\right] \leqslant \frac{4}{m} \qquad \text{(by Chebyshev's inequality)}$$

$$\leqslant \frac{1}{2}. \qquad \text{(because } m = \lfloor \frac{n}{2W} \rfloor \text{ and } n \geqslant 16W\text{)}$$

Let $\mathcal{G}$ be the event that $Y \geqslant \frac{m}{4}$. Then we have:

$$\mathbb{E}\left[R_T\right] = \mathbb{E}\left[R_T \mid \mathcal{G}\right] \cdot \mathbf{Pr}\left[\mathcal{G}\right] + \mathbb{E}\left[R_T \mid \neg\mathcal{G}\right] \cdot \mathbf{Pr}\left[\neg\mathcal{G}\right] \geqslant \mathbb{E}\left[R_T \mid \mathcal{G}\right] \cdot \frac{1}{2}.$$

Since at least $\frac{m}{4}$ of the events $\mathcal{F}_k$ occur when $\mathcal{G}$ occurs, and each event $\mathcal{F}_k$ induces at least $\frac{1}{2W} \cdot \frac{T}{n-W+1}$ regret, we have:

$$\mathbb{E}\left[R_T \mid \mathcal{G}\right] \geqslant \frac{m}{4} \cdot \frac{1}{2W} \cdot \frac{T}{n-W+1}.$$

Hence, we find:

$$\mathbb{E}\left[R_T\right] \geqslant \frac{m}{4} \cdot \frac{1}{2W} \cdot \frac{T}{n-W+1} \cdot \frac{1}{2} \geqslant \frac{m \cdot T}{16 \cdot W \cdot n}$$

$$\geqslant \frac{n}{4W} \cdot \frac{T}{16 \cdot W \cdot n} \qquad \text{(because } m = \lfloor \frac{n}{2W} \rfloor \geqslant \frac{n}{4W}\text{)}$$

$$= \frac{T}{64 \cdot W^2}.$$

For the upper bound, we proceed by running epoch-wise UCB using the $W$ memory size. There are algorithms, such as INF (Audibert & Bubeck, 2009), that can achieve a total regret of $O(\sqrt{nT})$ in a centralized setting with $n$ arms. For each window $j$ with $T_j$ number of pulls, we can utilize such an algorithm as a black box to achieve $O(\sqrt{WT_j})$ regret. Consequently, over the $n - W + 1$ windows, the overall total regret will be $O(\sum_{j=1}^{n-W+1} \sqrt{WT_j})$.

The regret is minimized when we concentrate all the pulls to *a single* window, i.e., we achieve $O(\sqrt{WT})$ regret. On the other hand, we can upper bound the regret by a simple application of the Cauchy–Schwarz inequality (using the constraint $(\sum_{j=1}^{n-W+1} T_j = T)$ as follows. We let $\boldsymbol{r}$ be the $(n - W + 1)$-dimensional vector that encodes all the $\sqrt{WT_j}$ for all $j$ and $\mathbf{1}$ as the all one vector of $(n - W + 1)$ dimension. We have

$$\sum_{j=1}^{n-W+1} \sqrt{WT_j} = \mathbf{1}^T \boldsymbol{r}$$

$$\leqslant \|\mathbf{1}\|_2 \|\boldsymbol{r}\|_2 \qquad \text{(by Cauchy–Schwarz)}$$

$$= \sqrt{n-W+1} \cdot \sqrt{\sum_{j=1}^{n-W+1} WT_j}$$

$$= \sqrt{W \cdot (n-W+1) \cdot T}.$$

Therefore, bringing back the leading constant, the total regret can be expressed as $O(\sqrt{W \cdot (n - W + 1) \cdot T})$, as desired.
Theorem 3 $\square$

# D Regret Minimization with an Everlasting Best Arm

Our main regret notion is based on the *epoch-wise* regret. However, in some practical scenarios, there are cases where the most popular item is not limited by the time horizon. Consider, again, the task of recommending movies to users for entertainment companies. On average, a movie remains in theaters for about 1 to 2 months. However, some exceptionally popular pieces can have a much longer run. For example, *The Sound of Music* was screened in theaters for 4 years and 6 months, while *Avatar* stayed for 34 weeks. Therefore, when designing a recommendation system for currently showing movies, we can assume a sliding window of 2 months, but there are some enduring ones that remain popular even after this window has passed. Similar situations occur in other contexts as well. For instance, the song *Lose Control* set a new record by spending 107 weeks on the Billboard Hot 100 chart, whereas the average lifespan of a song on the chart is typically between 6 and 7 weeks.

Inspired by these applications, we also propose another regret model, which we refer to as **regret minimization with an everlasting best arm**. In the scenario where there is an everlasting best arm, we can pull this best arm even if it appears more than $W$ time units earlier in the stream[6]. For situations involving such an everlasting best arm, there are two slightly different scenarios to consider: whether we are allowed to pull a sub-optimal expired arm and incur 1 regret, or simply receive a signal that a sub-optimal arm has expired and the sampling operation is disallowed.

## D.1 Regret minimization with everlasting best arm and explicit valid flag

The first scenario is when we cannot pull an expired arm other than the best arm. This means that the only arms available for pulling are the everlasting best arm and the $W$ most recent arms, and all expired arms (other than the best) in the memory will carry a flag of "being invalid" (or simply being deleted from the memory). For instance, if a movie is still being presented by the theater after the sliding-window period, it indicates that the movie has not expired and can still be selected. Therefore, in this setting, we can assume the existence of a flag for each arm indicating whether it is valid for pulling. Only the everlasting best arm and the $W$ most recent arms will have a positive flag.

**Definition 7** (Valid flag). Let $\{\texttt{arm}_i\}_{i=1}^n$ be a collection of $n$ arms with an everlasting best arm denoted as $\texttt{arm}^*$, and let $W$ be the window size. The valid flag is a function $\texttt{flag}(\texttt{arm}_i, t)$ that returns $\texttt{True}$ if $\texttt{arm}_i$ is $\texttt{arm}^*$ or if $i \geqslant t - W + 1$ (indicating that $\texttt{arm}_i$ is one of the $W$ most recent arms); it returns $\texttt{False}$ otherwise.

In simpler terms, $\texttt{flag}(\texttt{arm}_i, t) = \texttt{True}$ if and only if $\texttt{arm}_i$ is valid at time $t$. An equivalent definition is that the arm has not been forcibly deleted from the memory (by the adversary).

Next, we define regret minimization with an everlasting best arm and an explicit valid flag. In this setting, the $\texttt{flag}$ function is accessible to the algorithm, allowing it to determine whether an arm is valid without needing to pull it. This aligns with scenarios such as recommending movies that are currently showing, where we can ascertain whether a movie is still valid (i.e., still being presented in the theater) without any action.

**Definition 8** (Regret minimization with everlasting best arm and explicit valid flag). Let $\{\texttt{arm}_i\}_{i=1}^n$ represent a collection of $n$ arms with an everlasting best arm, $\texttt{arm}^*$. Let $W$ be the window size and $T$ be the total number of trials. Denote $\mu^*$ as the mean reward of the best arm $\texttt{arm}^*$ (among all arms), and let $t$ denote the variable for the index of the arriving arm. In this scenario, there exists an explicit $\texttt{flag}$ function, and an arm $\texttt{arm}_i$ can be pulled at time $t$ only if $\texttt{flag}(\texttt{arm}_i, t) = \texttt{True}$. Let $\{i(\tau)\}_{\tau=1}^T$ be the set of indices of arms pulled by some algorithm, and the regret is defined as $R_T := \sum_{\tau=1}^T (\mu^* - \mu_{i(\tau)})$.

If we have $W$ memory, a straightforward algorithm is to *not* pull any arm until $W$ steps and check whether the arm is still valid. The arm is valid if and only if it is the best arm, and we could therefore commit to the arm to achieve 0 regret. [7] As such, a natural question is whether we could do better with $o(W)$ memory. In what follows, we will show that the answer to the above question is *negative*: we will show that $\Omega(W)$ space is necessary to achieve a total regret of $o(T)$, essentially

---

[6]Note that this is different from the rest of the paper, where we require the expired arm to be deleted immediately from the memory.

[7]In this scenario, we assume that time continues to pass even when there are no more input arms available. Therefore, every arm, except for the everlasting best arm— including arm $\texttt{arm}_n$—may eventually expire as time goes on. Thus, we can simply wait long enough and use the $\texttt{flag}$ function to identify the everlasting best arm, which will be the only valid arm remaining.

An alternative assumption is that time ceases to progress when there are no new arms in the stream. In this case, the final valid arms will consist of the everlasting best arm plus the last $W$ arms, which are $\{\texttt{arm}_{n-W+1}, \cdots, \texttt{arm}_n\}$. If the everlasting best arm is not included among the last $W$ arms, we can identify it directly. If it is among the last $W$ arms, the problem then shifts to a centralized regret minimization problem with those $W$ arms. This represents a combination of our initial setting and the centralized regret minimization framework, so we will forego further discussion of this assumption.

indicating that the $W$-memory algorithm is *optimal*.

**Theorem 4.** *For any given parameters $T$, $n$, and $W$ such that $T \geqslant n \geqslant 4W$, there exists a family of streaming stochastic multi-armed bandit instances such that any single-pass streaming algorithm designed for a sliding-window stream of length $n$ with a window size $W$ and a memory of $\frac{W}{8}$ arms must incur a total expected regret of at least*

$$\mathbb{E}\left[R_T\right] \geqslant \frac{T}{120}.$$

*Furthermore, there exists an algorithm that given a stream of $n$ arms and parameters $W$ and $T$, achieves $0$ regret with a memory of $W$ arms.*

Theorem 4 shows an extremely sharp "phase transition" for the memory-regret trade-off: with $o(W)$ memory, we have to suffer $\Omega(T)$ regret. On the other hand, if we slightly increase the memory to $W$, we could achieve $0$ total regret.

To prove Theorem 4 we will utilize the following result on the sample-memory trade-offs for *storing an arm* in the memory from (Assadi & Wang, 2022).

**Proposition D.1** ((Assadi & Wang, 2022), cf. (Chen et al., 2024)). *Consider the following distribution of $m$ arms.*

---

**DIST$(m, \sigma, \beta)$: A hard distribution with $m$ arms for trapping the best arm**

1. *An index $i^*$ sampled uniform at random from $[m]$.*

2. *For $i \neq i^*$, let the arms be with reward $\mu_i = \sigma$.*

3. *For $i = i^*$, let the arm be with reward $\mu_{i^*} = \sigma + \beta$.*

---

*Any algorithm that outputs (the indices of) $\frac{m}{8}$ arms that contain the best arm on DIST with a probability of at least $\frac{2}{3}$ has to use at least $\frac{1}{1200} \cdot \frac{m}{\beta^2}$ arm pulls.*

The intuition behind our proof is that it is crucial not to overlook the best arm in a stream of options. By utilizing the distribution DIST, we can construct scenarios where it requires a considerable number of pulls to identify the best arm. Additionally, we can create instances that include multiple distributions of DIST, making it challenging to determine the best arm.

In these scenarios, an algorithm that uses a large number of arm pulls on earlier arms risks the possibility that the best arm may appear later in the stream. If the algorithm has already made too many pulls on suboptimal arms, it will incur substantial regret. Conversely, if the algorithm decides to conserve its pulls and primarily engages with the later part of the stream, there is a risk that the best arm could arrive early on, leading the algorithm to miss it entirely. As a result, any algorithm faces the inherent risk of missing the best arm, which can lead to significant regret.

***The proof of Theorem 4.*** In the discussion by the start of Section D.1, we have already introduced the relatively simple algorithm that uses $W$ arm memory and achieves $0$ regret. We focus on proving the lower bound in the proof.

According to Yao's minimax principle (Yao, 1977), it is sufficient to establish the lower bound for deterministic algorithms over a challenging distribution of inputs.

We will first introduce the CONST distribution for clarity.

---

**CONST$(m, \sigma)$: A distribution with $m$ arms with the same means**

1. $\forall i \in [m], \mu_i = \sigma$.

---

Let $\beta = \min\{\sqrt{\frac{W}{1200T}}, \frac{1}{10}\}$. Consider the following distribution of $n$ arms.

---

**SIGNAL**$(n, W, \beta)$**: A hard distribution with $n$ arms for regret minimization with an everlasting best arm with expiration signal**

1. The first $W$ arms of SIGNAL$(n, W, \beta)$ are DIST$(W, \frac{3}{5}, \beta)$.

2. The $W + 1$-th to $2W$-th arms are CONST$(W, \frac{1}{5})$.

3. With probability of $\frac{1}{2}$, $2W + 1$-th to $3W$-th arms are DIST$(W, \frac{2}{5}, \beta)$, otherwise, DIST$(W, \frac{4}{5}, \beta)$.

4. The remaining $n - 3W$ arms are CONST$(n - 3W, \frac{1}{5})$.

---

For any deterministic algorithm ALG, let $T_{\text{ALG}}$ denote the number of pulls it uses until time $2W$. Since ALG is a deterministic algorithm and the first $2W$ arms are the same in both scenarios, $T_{\text{ALG}}$ remains consistent regardless of whether the arms from $2W + 1$ to $3W$ follow the distribution DIST$(W, \frac{2}{5}, \beta)$ or DIST$(W, \frac{4}{5}, \beta)$.

Let $\mathcal{E}_1$ be the event that the arms from $2W + 1$ to $3W$ are DIST$(W, \frac{2}{5}, \beta)$ and $\mathcal{E}_2$ be the event that they are DIST$(W, \frac{4}{5}, \beta)$. If $T_{\text{ALG}} \leqslant \frac{T}{2}$, then we have:

$$\mathbb{E}\left[R_T\right] = \mathbb{E}\left[R_T | \mathcal{E}_1\right] \cdot \mathbf{Pr}\left[\mathcal{E}_1\right] + \mathbb{E}\left[R_T | \mathcal{E}_2\right] \cdot \mathbf{Pr}\left[\mathcal{E}_2\right] \geqslant \mathbb{E}\left[R_T | \mathcal{E}_1\right] \cdot \mathbf{Pr}\left[\mathcal{E}_1\right].$$

Since $\frac{1}{1200} \cdot \frac{W}{\beta^2} \geqslant T > \frac{1}{2}T \geqslant T_{\text{ALG}}$, by Proposition D.1, the probability that the best arm is stored in memory at time $W + 1$ is at most $\frac{2}{3}$. Let $\mathcal{F}$ be the event that the best arm is stored in memory at time $W + 1$. Then,

$$\mathbb{E}\left[R_T \mid \mathcal{E}_1\right] = \mathbb{E}\left[R_T \mid \mathcal{E}_1 \cap \mathcal{F}\right] \cdot \mathbf{Pr}\left[\mathcal{F}\right] + \mathbb{E}\left[R_T \mid \mathcal{E}_1 \cap \neg\mathcal{F}\right] \cdot \mathbf{Pr}\left[\neg\mathcal{F}\right] \geqslant \mathbb{E}\left[R_T \mid \mathcal{E}_1 \cap \neg\mathcal{F}\right] \cdot \mathbf{Pr}\left[\neg\mathcal{F}\right].$$

When the event $\mathcal{E}_1 \cap \neg\mathcal{F}$ occurs, the best arm is not stored in memory at time $2W + 1$, and all the arms with means $\frac{3}{5}$ have expired. The remaining arms have means at most $\frac{2}{5} + \beta \leqslant \frac{2}{5} + \frac{1}{10} = \frac{1}{2}$. Since we can only pull valid arms, the arms we can pull have a mean reward of at most $\frac{1}{2}$. Thus, the regret for each pull after time $2W$ is at least

$$\frac{3}{5} + \beta - \frac{1}{2} \geqslant \frac{1}{10}.$$

Therefore, we have:

$$\mathbb{E}\left[R_T \mid \mathcal{E}_1 \cap \neg\mathcal{F}\right] \geqslant (T - T_{\text{ALG}}) \cdot \frac{1}{10} \geqslant \left(T - \frac{T}{2}\right) \cdot \frac{1}{10} = \frac{T}{20}.$$

Thus,

$$\mathbb{E}\left[R_T\right] \geqslant \mathbb{E}\left[R_T \mid \mathcal{E}_1\right] \cdot \mathbf{Pr}\left[\mathcal{E}_1\right] \geqslant \mathbb{E}\left[R_T \mid \mathcal{E}_1 \cap \neg\mathcal{F}\right] \cdot \mathbf{Pr}\left[\neg\mathcal{F}\right] \cdot \mathbf{Pr}\left[\mathcal{E}_1\right] \geqslant \frac{T}{20} \cdot \frac{1}{3} \cdot \frac{1}{2} = \frac{T}{120}.$$

On the other hand, if $T_{\text{ALG}} \geqslant \frac{T}{2}$, then we have:

$$\mathbb{E}\left[R_T\right] \geqslant \mathbb{E}\left[R_T \mid \mathcal{E}_2\right] \cdot \mathbf{Pr}\left[\mathcal{E}_2\right].$$

When event $\mathcal{E}_2$ occurs, the best arm has a mean of $\frac{4}{5} + \beta$. The regret for each pull on the first $2W$ arms would be at least

$$\frac{4}{5} + \beta - \frac{3}{5} - \beta = \frac{1}{5}.$$

Therefore,

$$\mathbb{E}\left[R_T\right] \geqslant \frac{1}{2} \cdot \mathbb{E}\left[R_T \mid \mathcal{E}_2\right] \geqslant \frac{1}{2} T_{\text{ALG}} \cdot \frac{1}{5} \geqslant \frac{T}{20}.$$

In conclusion, since $\mathbb{E}\left[R_T\right] \geqslant \frac{T}{120}$ regardless of whether $T_{\text{ALG}} \geqslant \frac{T}{2}$ or not, we have completed our proof. Theorem 4 □

## D.2 Regret minimization with everlasting best arm and implicit valid flag

The second scenario is when we can still pull an expired arm, but doing so incurs a significant penalty. In this situation, any arm can be pulled at any time; however, if an expired arm is selected, a regret penalty of 1 is incurred.

Furthermore, we assume that the penalty associated with pulling an expired arm is not immediately known. If we were to be instantly informed about any penalties, the scenario could be simplified: we would incur a 1 regret penalty for all non-best arms in an effort to identify the best arm. This would lead to a total regret of $n - 1$, which is negligible given that $T \gg n$.

In this context, we can still assume the existence of a valid flag function, denoted as flag, to indicate whether an arm is valid. However, the algorithm cannot access this flag; therefore, it cannot determine whether an arm is valid.

**Definition 9** (Regret minimization with an everlasting best arm and implicit valid flag). Let $\{\texttt{arm}_i\}_{i=1}^n$ represent a collection of $n$ arms with an everlasting best arm, $\texttt{arm}^*$. Let $W$ be the window size and $T$ be the total number of trials. Denote $\mu^*$ as the mean reward of the best arm $\texttt{arm}^*$ (among all arms), and let $t$ denote the variable for the index of the arriving arm. In this scenario, an implicit flag function exists. Any arm $\texttt{arm}_i$ can be pulled at any time $t$, but if the flag indicates it is invalid (i.e., $\texttt{flag}(\texttt{arm}_i, t) = \texttt{False}$), a regret of 1 is incurred. Let $\{i(\tau)\}_{\tau=1}^T$ be the set of indices of arms pulled by a given algorithm, and let $\{\texttt{flag}_\tau = \texttt{flag}(\texttt{arm}_{i(\tau)}, t)\}_{\tau=1}^T$ represent the validity flag for the arms that were pulled. The total regret is defined as $R_T := \sum_{\texttt{flag}_\tau = \texttt{True}} (\mu^* - \mu_{i(\tau)}) + \sum_{\texttt{flag}_\tau = \texttt{False}} 1$.

In this setting, an $\Omega(W)$ memory is still necessary to achieve a total regret of $o(T)$. Although there is an everlasting best arm, the lack of a signal about whether an arm has expired makes this setting strictly more challenging than the one in which we receive an expiry signal. Thus, the claims and proofs applicable to the setting with signals also remain valid in this case.

**Lemma D.1.** *There exists a family of streaming stochastic multi-armed bandit instances such that, for any given parameters $T$, $n$, and $W$, where $T \geqslant n \geqslant 4W$, any single-pass streaming algorithm for a sliding-window stream of length $n$ with a window size $W$ and a memory of $\frac{W}{8}$ arms must incur a total expected regret of at least*

$$\mathbb{E}\left[R_T\right] \geqslant \frac{T}{120}.$$

However, in this setting, the upper bound of total regret with trivial space complexity $W - 1$ is no longer $O_{n,W}(1)$. Since we are not informed whether an arm is expired, it becomes challenging to identify the best arm easily. When arms have means close to the best arm, we must pull these arms numerous times, which leads to significant regret, as each pull on an expired arm incurs a regret penalty of 1. Furthermore, even after many pulls on these arms, we may still incorrectly identify the best arm, resulting in additional regret. In fact, achieving $o(T)$ total regret is impossible even with a memory capacity of $n - 1$, which is an even stronger condition than $W - 1$ memory.

**Lemma D.2.** *There exists a family of streaming stochastic multi-armed bandit instances such that, for any given parameters $T$, $n$, and $W$ with $T \geqslant n \geqslant 8W$, any single-pass streaming algorithm for a sliding window stream of length $n$, with a window size $W$ and a memory of $n - 1$ arms, must incur*

$$\mathbb{E}[R_T] \geqslant \frac{T}{1800}$$

*total expected regret.*

The intuition behind this is that we cannot identify the best arm even after $T$ pulls in $\text{DIST}(W, \frac{3}{5}, \beta)$ with $\beta = O\left(\frac{1}{T}\right)$, since $O\left(\frac{W}{\varepsilon^2}\right)$ pulls are required to identify the $\varepsilon$-best arm among $W$ arms. In this scenario, when the arm is not expired, it will incur $\beta$ regret per pull, while an expired arm incurs a regret of 1. Thus, a sound strategy would be to pull the arms before they expire. However, if there are two such distributions in the stream, we cannot determine in advance which distribution contains the best arm. Consequently, we risk either overspending pulls on the wrong distribution or incurring 1 regret per pull by not allocating most pulls to the correct distribution. This results in a total regret of $\Theta(T)$ in either situation.

*Proof.* By Yao's minimax principle (Yao, 1977), it is sufficient to demonstrate the lower bound for deterministic algorithms in the face of a challenging distribution of inputs. Let's consider the following distribution:

---

**SIGNAL$'(n, W, \beta)$: A hard distribution with $n$ arms for regret minimization with an everlasting best arm with expiration signal**

1. The first $W$ arms of SIGNAL$(n, W, \beta)$ is DIST$(W, \frac{3}{5}, \beta)$.

2. The $W + 1$-th to $2W$-th arms are CONST$(W, \frac{1}{5})$.

3. With probability of $\frac{1}{2}$, $2W + 1$-th to $3W$-th arms are DIST$(W, \frac{2}{5}, \beta)$, otherwise, DIST$(W, \frac{4}{5}, \beta)$.

4. The remaining $n - 3W$ arms are CONST$(n - 3W, \frac{1}{5})$.

---

According to Proposition B.1, there exist constants $c_1$ and $c_2$ such that any algorithm using no more than $c_1 \frac{W}{\beta^2} \log c_2$ pulls will not reliably return the $\frac{\beta}{2}$-best arm with a probability of at least $\frac{3}{4}$. We set $\beta = \min\{\frac{1}{10}, \frac{c_1 W}{2T} \log c_2\}$.

Now, consider SIGNAL$'(n, W, \beta)$. Let $T_1$ denote the number of pulls made before time $2W + 1$, $T_2$ the number of pulls made between times $2W + 1$ and $4W$, and $T_3$ the number of pulls made after time $4W$. Let $\mathcal{E}_i$ represent the event that $T_i \geqslant \frac{T}{3}$. Since $T_1 + T_2 + T_3 = T$, at least one of the events $\mathcal{E}_i$ must occur. Thus, we have:

$$\mathbb{E}[R_T] = \mathbb{E}[R_T \mid \mathcal{E}_i] \cdot \mathbf{Pr}[\mathcal{E}_i] + \mathbb{E}[R_T \mid \neg\mathcal{E}_i] \cdot \mathbf{Pr}[\neg\mathcal{E}_i] \geqslant \mathbb{E}[R_T \mid \mathcal{E}_i] \cdot \mathbf{Pr}[\mathcal{E}_i].$$

Therefore, it suffices to show that $\mathbb{E}[R_T \mid \mathcal{E}_i] \geqslant \frac{T}{600}$ for each $i \in [3]$.

**Case 1: $\mathcal{E}_1$ occurs.** Let $\mathcal{F}_1$ be the event that the arms from $2W + 1$ to $3W$ are drawn from DIST$(W, \frac{2}{5}, \beta)$, and let $\mathcal{F}_2$ be the event that these arms are drawn from DIST$(W, \frac{4}{5}, \beta)$. Hence,

$$\mathbb{E}[R_T \mid \mathcal{E}_1] \geqslant \mathbb{E}[R_T \mid \mathcal{E}_1 \cap \mathcal{F}_2] \cdot \mathbf{Pr}[\mathcal{F}_2] = \mathbb{E}[R_T \mid \mathcal{E}_1 \cap \mathcal{F}_2] \cdot \frac{1}{2}.$$

When $\mathcal{F}_2$ occurs, each pull on the first $2W$ arms results in at least $\frac{1}{10}$ regret. Since we spend at least $\frac{T}{3}$ pulls on these first $2W$ arms when $\mathcal{E}_1$ occurs, we have:

$$\mathbb{E}[R_T \mid \mathcal{E}_1] \geqslant \mathbb{E}[R_T \mid \mathcal{E}_1 \cap \mathcal{F}_2] \cdot \frac{1}{2} \geqslant \frac{1}{10} \cdot \frac{T}{3} \cdot \frac{1}{2} = \frac{T}{60}.$$

**Case 2: $\mathcal{E}_2$ occurs.** Similarly,

$$\mathbb{E}[R_T \mid \mathcal{E}_2] \geqslant \mathbb{E}[R_T \mid \mathcal{E}_2 \cap \mathcal{F}_1] \cdot \mathbf{Pr}[\mathcal{F}_1] = \mathbb{E}[R_T \mid \mathcal{E}_2 \cap \mathcal{F}_1] \cdot \frac{1}{2}.$$

When $\mathcal{F}_1$ occurs, each pull on the arms from $2W + 1$ to $4W$ leads to at least $\frac{1}{10}$ regret. Since we make at least $\frac{T}{3}$ pulls in this range when $\mathcal{E}_2$ occurs,

$$\mathbb{E}[R_T \mid \mathcal{E}_2] \geqslant \mathbb{E}[R_T \mid \mathcal{E}_2 \cap \mathcal{F}_1] \cdot \frac{1}{2} \geqslant \frac{1}{10} \cdot \frac{T}{3} \cdot \frac{1}{2} = \frac{T}{60}.$$

**Case 3: $\mathcal{E}_3$ occurs.** Given that $\beta = \frac{c_1 W}{2T} \log c_2$, it is impossible to return the $\frac{\beta}{2}$-best arm (which is the exact best arm in this distribution) with a probability of at least $\frac{3}{4}$ by using a maximum of $T$ pulls. Let $\mathcal{G}$ be the event that we pull at most $\frac{T}{6}$ times on the best arm after time $4W$. If $\mathbf{Pr}[\mathcal{G}] \leqslant \frac{1}{10}$, we can devise a strategy that distinguishes the best arm from the others, which is impossible. Hence, $\mathbf{Pr}[\mathcal{G}] \geqslant \frac{1}{10}$.

After time $4W$, pulling a valid arm with a mean of $\frac{1}{5}$ results in at least $\frac{1}{10}$ regret, while pulling from an expired arm incurs a regret of 1. Thus, we incur at least $\frac{1}{10}$ regret if we do not pull the best arm after time $4W$. When $\mathcal{G}$ occurs, we will spend at least $\frac{T}{6}$ pulls on arms other than the best arm beyond time $4W$, leading to:

$$\mathbb{E}[R_T \mid \mathcal{E}_3] \geqslant \mathbb{E}[R_T \mid \mathcal{E}_3 \cap \mathcal{G}] \cdot \mathbf{Pr}[\mathcal{G}] \geqslant \frac{1}{10} \cdot \frac{T}{6} \cdot \frac{1}{10} = \frac{T}{600}.$$

Since at least one of the events $\mathcal{E}_i$ must occur, we have:

$$\max_{i \in [3]} \{\mathbf{Pr}\left[\mathcal{E}_i\right]\} \geqslant \frac{1}{3}.$$

Therefore,

$$\mathbb{E}\left[R_T\right] \geqslant \max_{i \in [3]} \{\mathbb{E}\left[R_T \mid \mathcal{E}_i\right] \cdot \mathbf{Pr}\left[\mathcal{E}_i\right]\} \geqslant \frac{T}{600} \cdot \max_{i \in [3]} \{\mathbf{Pr}\left[\mathcal{E}_i\right]\} \geqslant \frac{T}{600} \cdot \frac{1}{3} = \frac{T}{1800}.$$

$\square$

## E   Failure of State-of-the-Art Algorithms in Vanilla Streaming MABs

In this section, we will provide simple counterexamples and proofs to illustrate why the state-of-the-art algorithms used in vanilla streaming multi-armed bandits (MABs) do not perform well in sliding-window MABs.

It is important to note that, based on Theorems 1 and 3 that we have established, we have demonstrated that any single-pass streaming algorithm utilizing $o(W)$ memory will fail to address both the weak and strong exploration problems or achieve a regret of $o(T)$. Consequently, the state-of-the-art algorithms from vanilla streaming MABs will also fail in the sliding-window setting, as supported by our theorems. While we have already established this in a general context, we believe that providing a simpler, specific proof related to these algorithms will help readers better understand why algorithms with $o(W)$ memory fail in sliding-window scenarios.

The state-of-the-art algorithm for streaming exploration was proposed by Assadi & Wang (2020). This algorithm can identify the best arm with a probability of at least $1 - \delta$ using only 1 unit of memory and $O\left(\frac{n}{\varepsilon^2} \log\left(\frac{1}{\delta}\right) + \log^2(n) \cdot \frac{\log^2\left(\frac{1}{\delta}\right)}{\varepsilon^3}\right)$ pulls.

However, we will present a counterexample demonstrating that the algorithm by Assadi & Wang (2020) cannot effectively address weak exploration in the sliding-window setting, even with a success probability of at least $0.6$.

**Lemma E.1.** *There exists a data stream such that the algorithm by Assadi & Wang (2020) cannot resolve the weak exploration for this stream with a probability of at least $0.6$.*

*Proof.* Consider the data stream defined by $\mu_i = 1 - \frac{i-1}{n}$, which has a decreasing mean. In this scenario, the best arm from time $t = 1$ to $W$ is $\texttt{arm}_1$, while the best arm at time $t = i$ for $i > W$ is $\texttt{arm}_{i-W+1}$.

If the algorithm can correctly identify the best arm from time $t = 1$ to $W$ with a probability of at least $0.6$, since it has only 1 unit of memory, the arm stored at that time must be $\texttt{arm}_1$. Consequently, at time $t = W$, only $\texttt{arm}_1$ will be stored in memory, causing us to lose access to $\texttt{arm}_2$ indefinitely, which is the best arm at time $t = W + 1$. Thus, it becomes impossible for the algorithm to return the correct best arm at time $t = W + 1$ with a probability greater than $1 - 0.6 = 0.4$. This indicates that the algorithm fails to resolve weak exploration with a probability of at least $0.6$.

Conversely, if the algorithm fails to return the correct best arm from time $t = 1$ to $W$ with a probability of at least $0.6$, it also indicates a failure in solving weak exploration with a probability of at least $0.6$.

Therefore, it is impossible for the algorithm to successfully resolve weak exploration with a probability of at least $0.6$.   $\square$

The state-of-the-art algorithm for regret minimization was proposed by Wang (2023). This algorithm achieves a total regret of $O\left(n^{1/3}T^{2/3}\right)$, which matches the lower bound of regret for streaming multi-armed bandits (MABs), using $\lceil \log^* n \rceil + 1$ units of memory. The key idea behind their algorithm is to first identify an $\varepsilon$-best arm for the entire stream and then allocate the remaining pulls to that $\varepsilon$-best arm. By choosing $\varepsilon = O\left(\sqrt[3]{\frac{n \log n}{T}}\right)$, this approach minimizes the total regret to $O\left(n^{1/3}T^{2/3}\right)$.

However, their algorithm cannot be applied to the sliding-window setting. The strategy of locating an $\varepsilon$-best arm and dedicating all remaining pulls to it is flawed, as the $\varepsilon$-best arm can expire over time, making it unavailable for pulling when it does. Therefore, the algorithm proposed by Wang (2023) is not even well-defined in the sliding-window context.

A natural attempt to adapt the algorithm for the sliding-window setting would be to identify an $\varepsilon$-best arm for each epoch and allocate all remaining pulls for that epoch to the identified arm. However, we will demonstrate that there exist data streams that require the algorithm to utilize at least $O\left(\frac{1}{\varepsilon}\right)$ memory to return the $\varepsilon$-best arm for each epoch.

**Lemma E.2.** *There exist data streams and $\varepsilon \in (0, 1)$ such that it is impossible to return the $\varepsilon$-best arm at any time $t$ with a probability of at least $0.6$ using only $o\left(\frac{1}{\varepsilon}\right)$ memory.*

*Proof.* Let $\varepsilon = \frac{1}{100 \cdot W}$. Define the means of the arms as follows:

$$\mu_i = 1 - 2\varepsilon \cdot \left(i - 1 - 50 \cdot W \cdot \left\lfloor \frac{i}{50 \cdot W} \right\rfloor\right).$$

This means the data stream consists of arms with decreasing means, where each subsequent arm has a mean $2\varepsilon$ smaller than the previous arm. The mean of an arm is reset to $1$ when it reaches $0$.

For this type of data stream, since each subsequent arm has a mean $2\varepsilon$ lower than its predecessor, the $\varepsilon$-best arm at any time $t$ is actually the best arm at that time. Consequently, the algorithm must store the best arm to accurately return the $\varepsilon$-best arm. Given the decreasing nature of the arm means, it is necessary to keep track of all arms in the window, requiring a memory size of $W = \Omega\left(\frac{1}{\varepsilon}\right)$. $\qquad\square$

Thus, since certain data streams exist where $W = o\left(\sqrt[3]{\frac{n \log n}{T}}\right)$, it follows that the adapted algorithm cannot operate with $o(W)$ memory for such streams.

# F    Additional Details and Results of the Experiments

We provide additional details and results of the experiments in different settings, including the experiments with real-world datasets. We start with the information of the real-world dataset.

**Real-world data and experiment settings.** For the real-world data, we utilize the MovieLens 1M dataset as referenced in (Harper et al., 2016). This dataset comprises over 1 million anonymous movie ratings, totaling 1,000,209 ratings from around 3900 films, contributed by 6040 users who joined MovieLens in the year 2000.

In our experiment, we treat each movie as an arm, with the reward for a given arm (movie) being its average rating. The ratings are rescaled and normalized to a range of $[0, 1]$. When we pull an arm (movie), we receive a random rating for that movie. For the $\varepsilon$-exploration experiments on the MovieLens 1M dataset, we use $n \in \{1000, 2000, 3500\}$ and $W \in \{20, 50, 100, 200\}$ (the dataset only has 3500 movies). For the regret minimization experiments, we use $n \in \{500, 1000\}$ and $W \in \{20, 50\}$.

## F.1    Experimental results on pure exploration

The experiments for the exploration algorithm with the synthetic and real-world datasets are shown in Figures 3 to 6 and in Figures 7 to 9.

From the figures, it could be observed that there are generally trade-offs between memory/quality. The trade-off curve for the memory/quality is mostly stable for our algorithm: for the mean and median statistics, the error bar obtained from $10$ runs is quite narrow. On the other hand, for the baseline algorithm with top-$k$ streaming exploration algorithm, increasing the memory almost gives *no improvement* for the performance.

The performance gap between the baseline and our algorithm is quite significant in the figures. The main reason for this gap is as follows: in vanilla streaming multi-armed bandits (MABs), there is no concept of expiration, which means that the design of these algorithms does not account for the possibility that all the arms stored in memory can expire at a certain point in time. If there is no valid arm stored in the memory, the identification task at that step simply fails and we charge the gap as the mean of the best arm in the window.

The advantages of our algorithms are consistent with both the synthetic and the real-world datasets. In the real-world dataset, the advantage of our algorithms is slightly less significant when the memory is very small. However, the moment we use a memory of, e.g., 10 arms, the advantage again becomes significant.

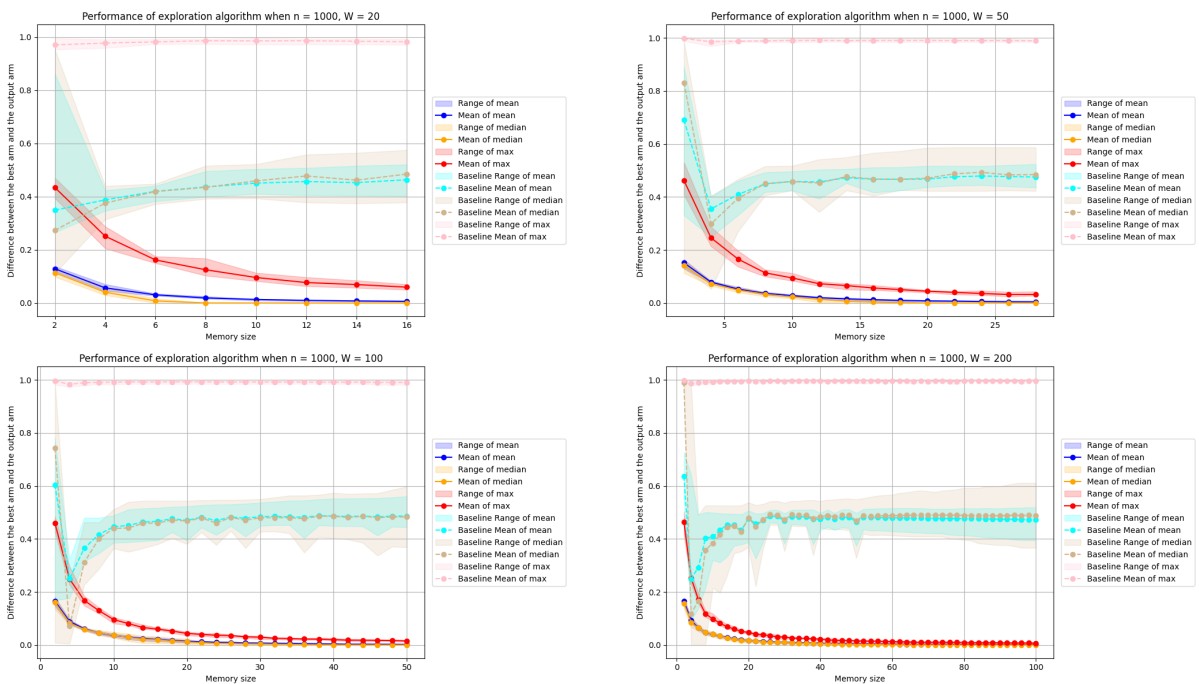

*Figure 3.* An illustration of the memory-quality trade-off in $\varepsilon$-exploration for $n = 1000$.

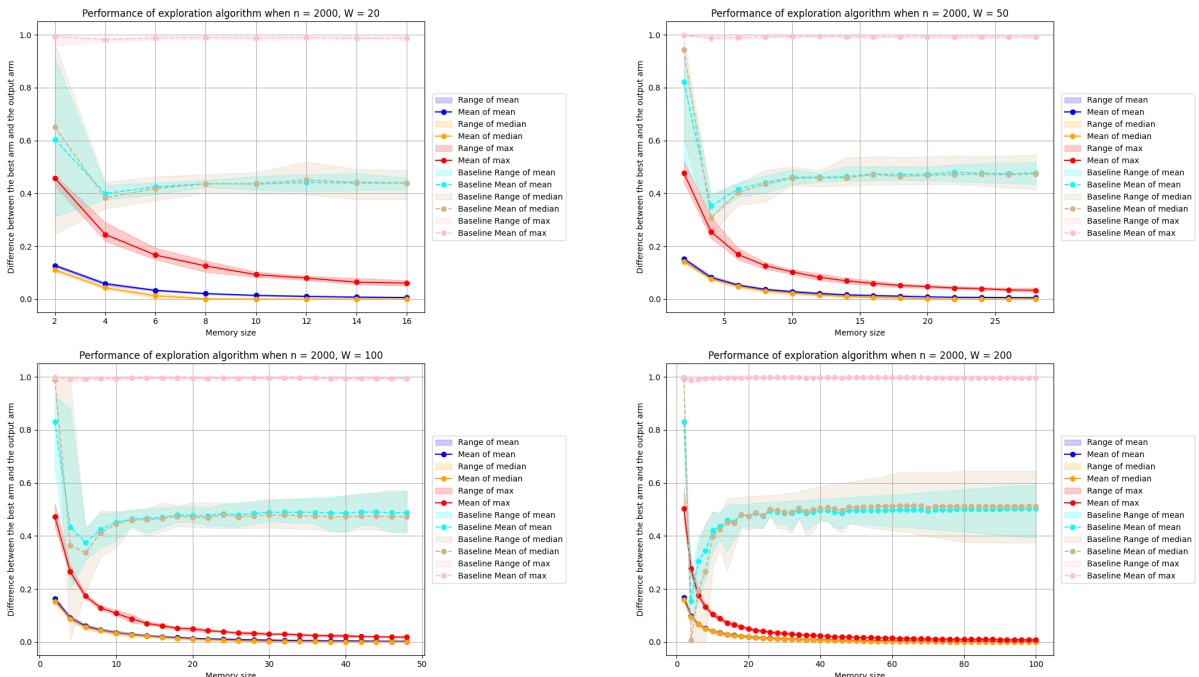

*Figure 4.* An illustration of the memory-quality trade-off in $\varepsilon$-exploration for $n = 2000$.

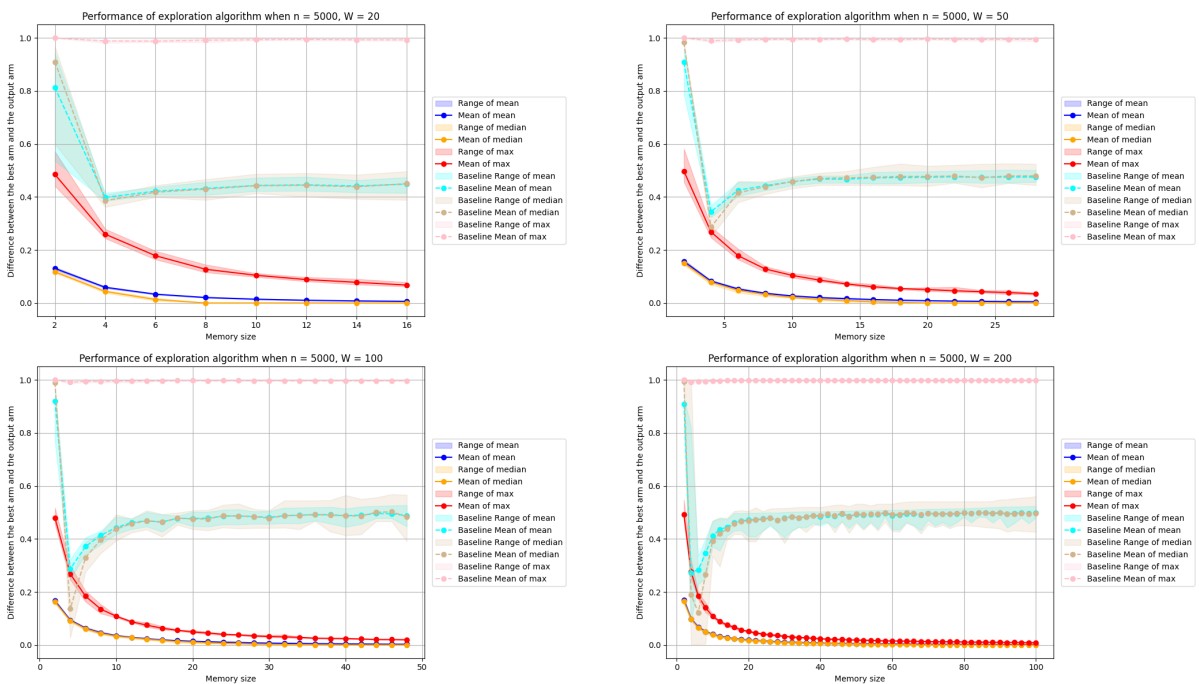

*Figure 5.* An illustration of the memory-quality trade-off in $\varepsilon$-exploration for $n = 5000$.

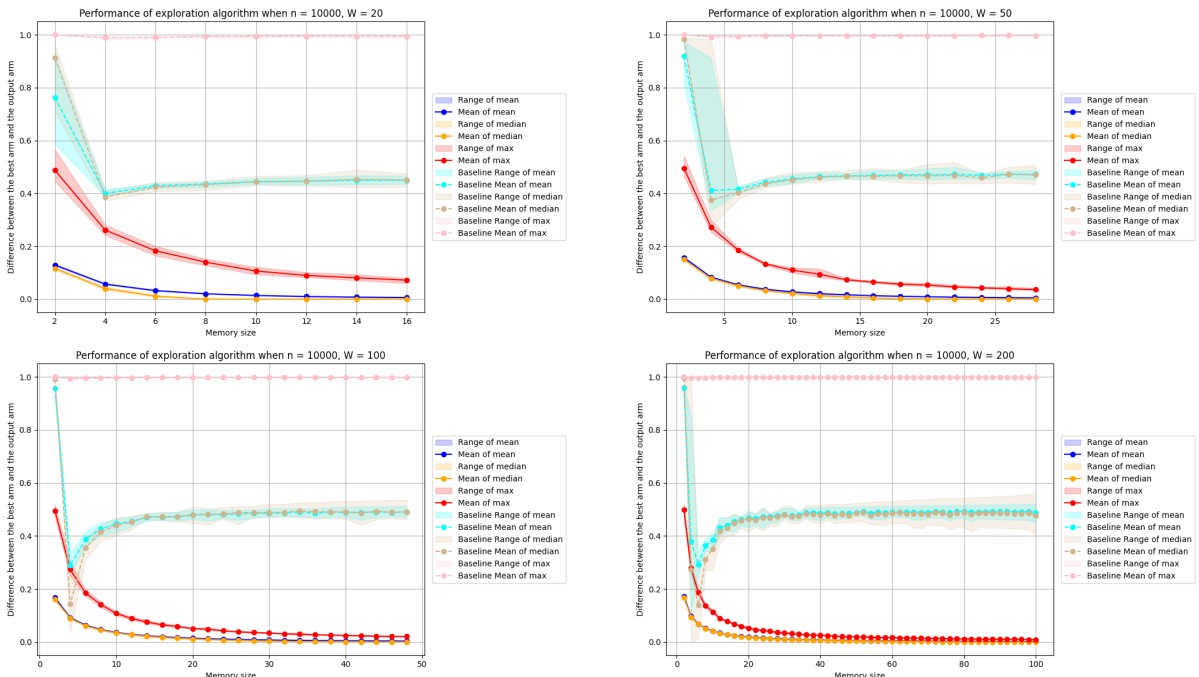

*Figure 6.* An illustration of the memory-quality trade-off in $\varepsilon$-exploration for $n = 10000$.

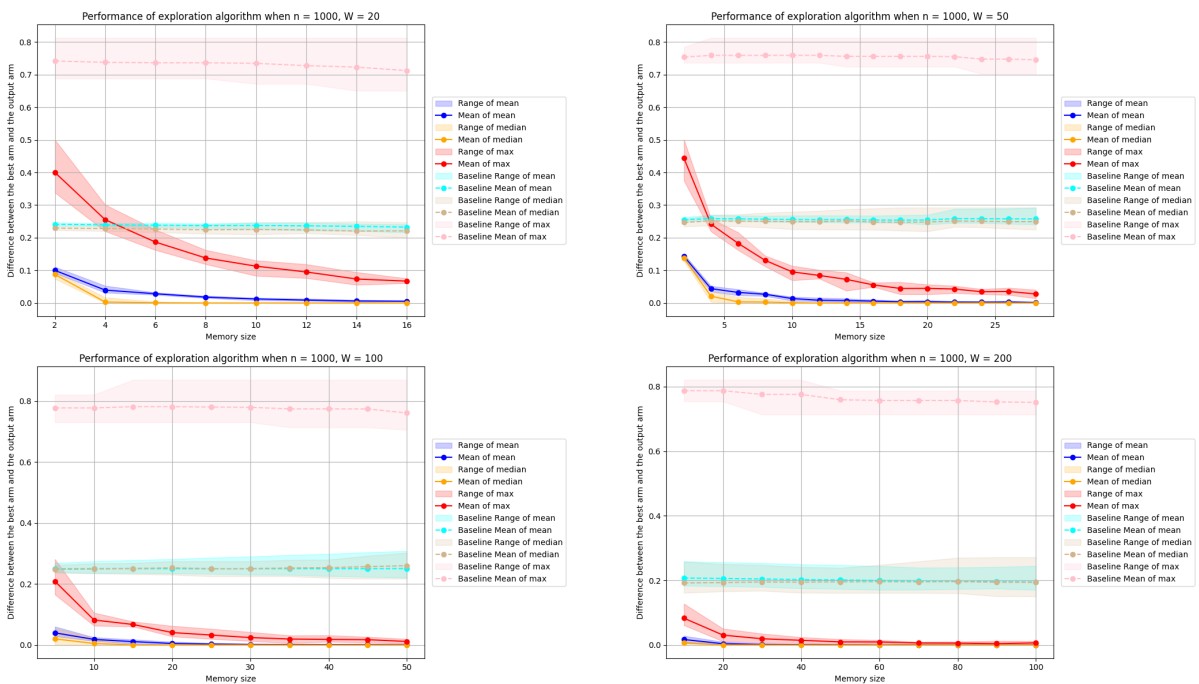

*Figure 7.* An illustration of the memory-quality trade-off in $\varepsilon$-exploration for $n = 1000$.

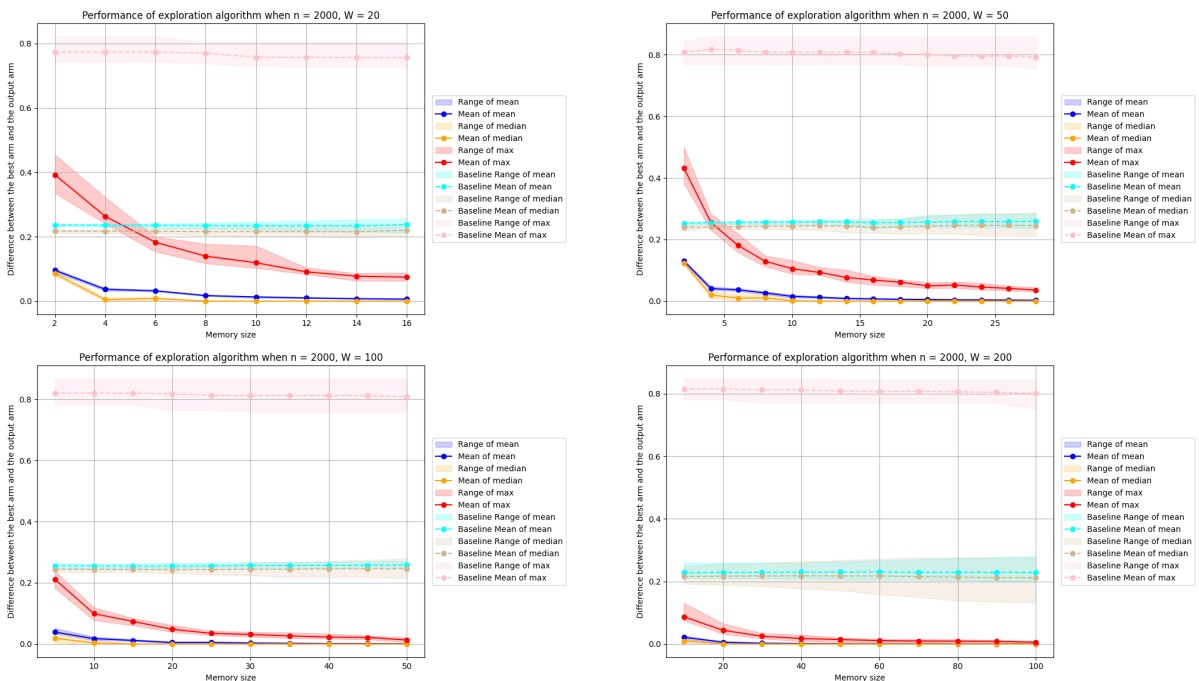

*Figure 8.* An illustration of the memory-quality trade-off in $\varepsilon$-exploration for $n = 2000$.

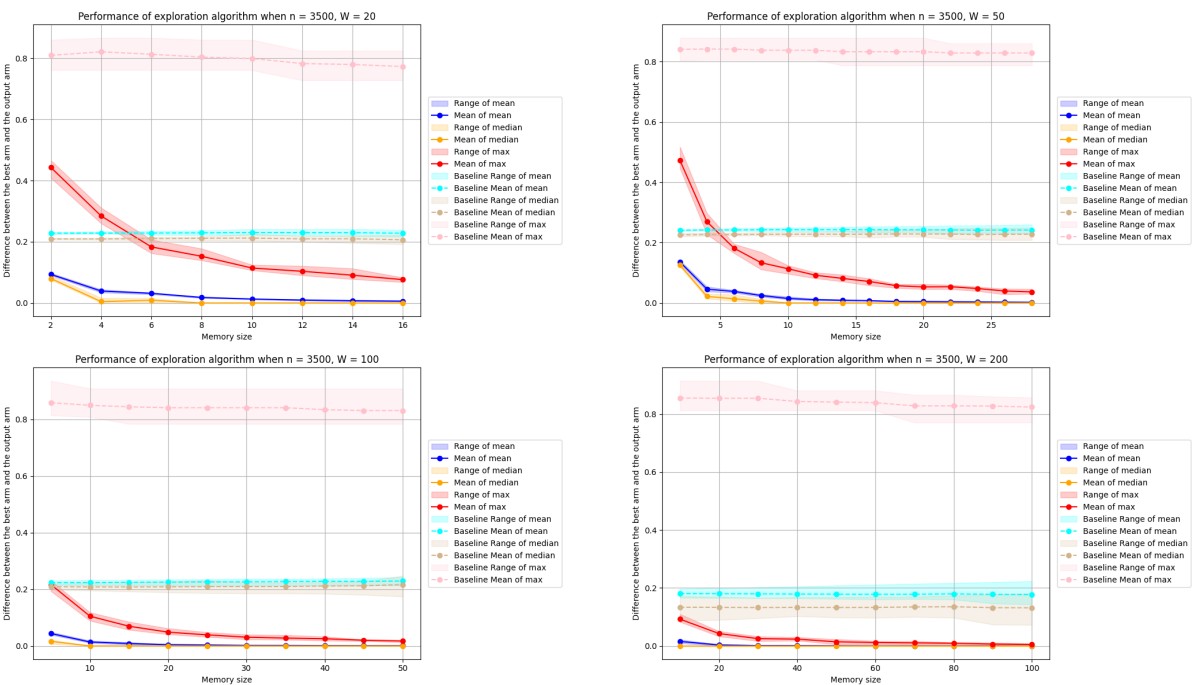

*Figure 9.* An illustration of the memory-quality trade-off in $\varepsilon$-exploration for $n = 3500$.

## F.2 Experimental results with epoch-wise regret minimization

The experimental results for regret minimization in the epoch-wise regret setting are shown as Figures 10 and 11. The budgets are *evenly distributed* over different epochs, i.e., $T_1 = T_2 = \cdots = T_{n-W+1} = \frac{T}{n-W+1}$. Again, with 10 independent runs of the algorithm, we report both the mean and the range of the regrets. We also report the mean and the range of the regrets for the baseline algorithm that performs exploration-then-commit.

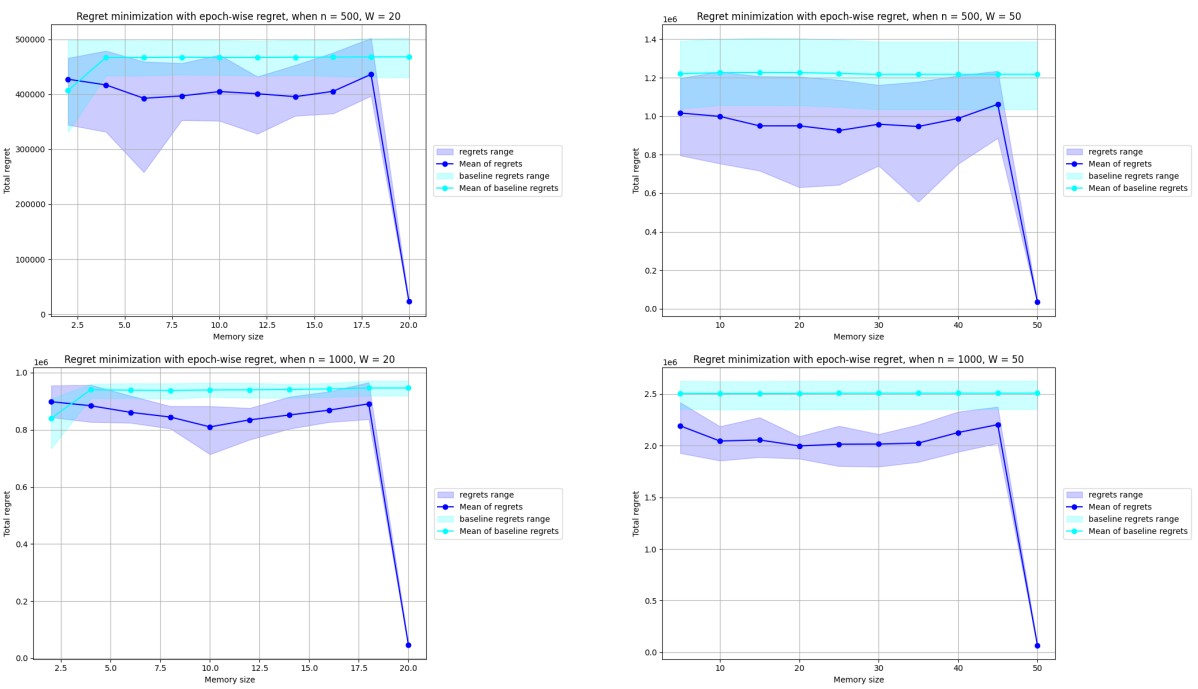

*Figure 10.* An illustration of the memory-quality trade-off in regret minimization for synthetic data.

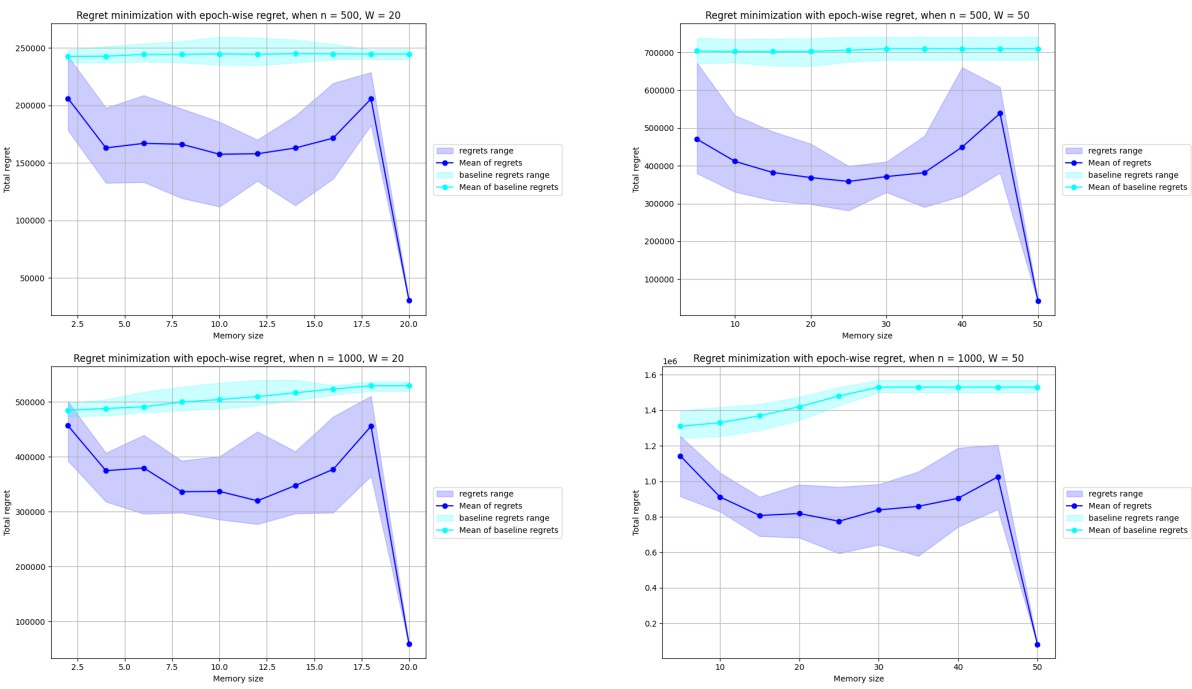

*Figure 11.* An illustration of the memory-quality trade-off in regret minimization for real-world data.

From the figures, it can be observed that our algorithm significantly outperforms the baseline algorithm. The regrets for the synthetic dataset follow a decreasing trend when the memory increases. Interestingly, the regrets slightly go up as the memory increases before the memory becomes $W$. We suspect this to be an artifact of the reservoir sampling process we relied on when the memory is lower than $W$.

### F.3 Experiments for regret with the everlasting best arm

The experimental results for regret minimization with the everlasting best arm are shown as Figures 12 to 14. Here, we commit to any arm in the end if we do not have the best arm in the memory. With 10 independent runs of the algorithm, we report both the mean regret and the range of the regrets.

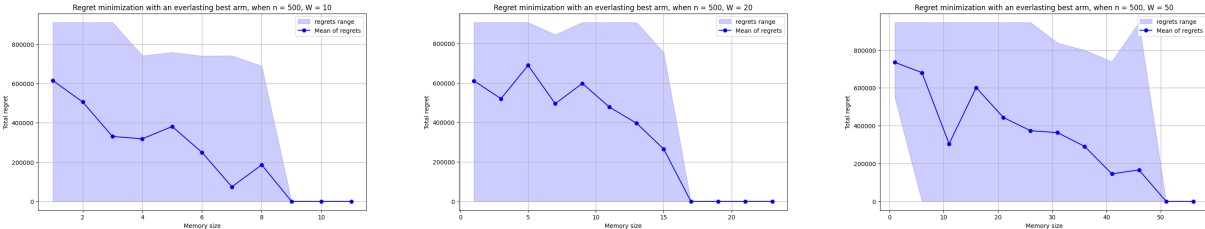

*Figure 12.* An illustration of the memory-regret trade-off for the everlasting best arm setting with $n = 500$.

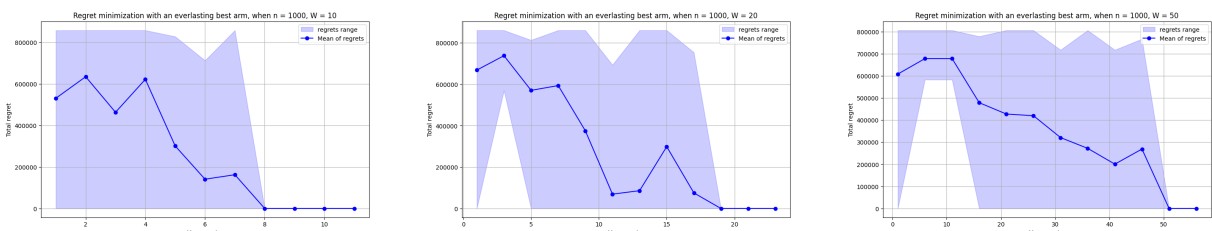

*Figure 13.* An illustration of the memory-regret trade-off for the everlasting best arm setting with $n = 1000$.

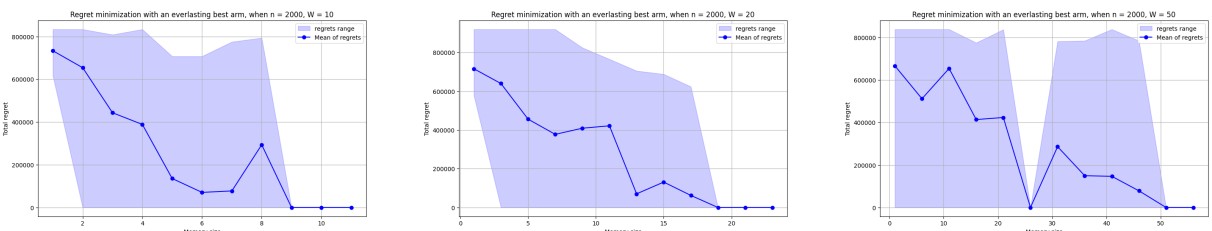

*Figure 14.* An illustration of the memory-regret trade-off for the everlasting best arm setting with $n = 2000$.

As we could observe in the figures, among the 10 executions of the algorithm, although the algorithm might get "lucky" with $o(W)$ memory, the range of the regret before reaching the $W$ memory is always wide, and the regret could always be high. On the other hand, after we have $W$ memory, we could easily identify the best arm and achieve 0 regret. The ranges observe a sharp drop at the $W$-memory point, which validates our theoretical results.

