# OpenReview forum: "Online Learning with Recency: Algorithms for Sliding-window Streaming Multi-armed Bandits"
_ICML.cc/2026/Conference — ICML 2026 regular_

### Official Review · Reviewer_JZ9W · 2026-02-25

**Soundness:** 3
**Presentation:** 2
**Significance:** 3
**Originality:** 2
**Overall Recommendation:** 4
**Confidence:** 2

**Summary:**

This paper studies a streaming multi-armed bandit setting with recency constraints: arms arrive sequentially in a single pass, and at each time $t$ only the most recent $W$ arms are valid. The authors formalize sliding-window streaming MABs and show that this model exhibits qualitatively different feasibility and memory trade-offs than standard bandits. For pure exploration, they distinguish weak vs strong guarantees and prove a sharp separation: exact best-arm identification essentially requires $\Omega(W)$ memory, while $\varepsilon$-best identification is achievable with only $O(1/\varepsilon)$ memory via a simple bucket scheme. For regret minimization, they introduce an epoch-wise regret notion to avoid coupling the benchmark to the algorithm’s interaction with the stream, and establish a memory–regret phase transition: with sub-$W$ memory one must incur $\Omega(T/W^2)$ regret on some instances, whereas with $\Theta(W)$ memory an algorithm can achieve $O(\sum_j \sqrt{W T_j})$ regret. Overall, the paper offers a clean model for recency-aware online learning and provides principled lower/upper bounds that clarify which goals are attainable under stringent streaming and memory constraints.

**Compliance With Llm Reviewing Policy:**

Affirmed.

**Final Justification:**

I will keep my score for  the reasons mentioned in my rebuttal acknowledgement.

**Key Questions For Authors:**

BUCKET(s) evaluates every arriving arm with a fixed budget $s$, which may be computationally wasteful in practice since most low-mean arms never affect the final recommendation. Is it possible to design an adaptive sampling procedure that allocates fewer samples to evidently suboptimal arms (and in the scenario where many high-reward buckets have been updated very recently) under the sliding-window streaming constraint? If not, what is the fundamental obstacle?

**Limitations:**

Yes

**Strengths And Weaknesses:**

**Strengths**

1. This paper formalizes a recency-aware streaming bandit model that naturally combines sliding-window streaming with streaming MABs, and the proposed BUCKET algorithm is simple and easy to implement.
2. The paper provides comprehensive theoretical results, establishing near-tight upper/lower bounds that characterize sharp memory–sample/regret trade-offs for both exact and $\epsilon$ exploration and epoch-wise regret minimization.
3. Numerical experiments align with the theory and empirically validate the predicted trade-offs/transition behavior under limited memory.



**Weaknesses**

1. The epoch-wise regret definition assumes a pre-specified, non-adaptive pull budget $\{T_j\}$ per epoch, which may be restrictive compared to some flexible sliding-window settings where the sampling effort can adapt to the observed difficulty or non-stationarity.
2. The algorithm requires the knowledge of $n$, however, in real-world streaming systems, $n$ is usually unknown or can be arbitrary large.


Suggestions:

1. In Algorithm 1, it is better to write the comment `// Expired arms also discarded` formally in the pseudocode.

---

> ### Author Rebuttal · Authors · 2026-03-31
>
> We thank the reviewer for the positive evaluation and helpful comments. Our responses are as follows.
>
> > The epoch-wise regret definition assumes a pre-specified, non-adaptive pull budget $T_j$ per epoch, which may be restrictive compared to some flexible sliding-window settings where the sampling effort can adapt to the observed difficulty or non-stationarity.
>
> Our algorithm does not require the value of $T_j$ until the $j$-th epoch. Whether $T_j$ is fixed before start or adaptively chosen by an adversary during execution, the algorithm maintains the same upper bound due to the guarantees for epoch-wise UCB.
>
> > The algorithm requires the knowledge of n, however, in real-world streaming systems, n is usually unknown or can be arbitrary large.
>
> The only algorithm in our paper that needs n is the algorithm that solves the strong $\varepsilon$-exploration. By definition, it guarantees that at each time $t \in \{ 1, 2, \cdots, n \}$, the algorithm selects an arm $\text{arm}_i$ such that, with probability at least $1-\delta$, all these selected arms are $\varepsilon$-best arms at their respective times. To ensure an overall success probability, we consider not just individual events but their union bound, which naturally introduces a dependency on $n$. Our weak $\varepsilon$-exploration and regret-minimization algorithms do not require the knowledge of $n$.
>
> > In Algorithm 1, it is better to write the comment // Expired arms also discarded formally in the pseudocode.
>
> Thank you for the writing suggestion. We will incorporate your suggestion to enhance the presentation in the revised version.
>
> > BUCKET(s) evaluates every arriving arm with a fixed budget s, which may be computationally wasteful in practice since most low-mean arms never affect the final recommendation. Is it possible to design an adaptive sampling procedure that allocates fewer samples to evidently suboptimal arms (and in the scenario where many high-reward buckets have been updated very recently) under the sliding-window streaming constraint? If not, what is the fundamental obstacle?
>
> This is indeed a compelling question. While the strategy does sound interesting to us, we can identify at least two main challenges for this idea:
>
> - Firstly, the risk of failure can “cascade” to future arms. If we reduce sampling for a suboptimal arm to save time, it might be placed in a bucket far from its true position. This adaptive strategy can cause problems over time. Specifically, a better arm might expire, allowing a less optimal arm—sampled less frequently—to be incorrectly identified as the best. Furthermore, once this happens, a later better arm, even if we allocate many samples, may still not be able to defeat the bad arm. In other words, once a failure case happens, we cannot have any control over the behavior of the algorithm.
>
> - Secondly, it is a bit hard to imagine that the more conservative strategy can reduce the *asymptotic* sample complexity for identifying the $\varepsilon$-best arm. Even in the offline setting, the worst-case complexity for identifying $\varepsilon$-best arm is still $\Omega(n/\varepsilon^2)$. Therefore, the proposed strategy will most likely only reduce the leading constant on the sample complexity.
>
> With the above being said, the reviewer’s comment presents a fascinating direction to pursue. Even if the asymptotic sample complexity remains the same, we might be able to save many “actual samples” in the experiments. We will leave this direction as an interesting future problem.

---

> > ### Author Rebuttal · Reviewer_JZ9W · 2026-04-02
> >
> > We thank the authors for the detailed response. This paper indeed proposes an interesting direction for future study. The rebuttal clarifies some of my concerns, especially regarding the role of $T_j$ in the epoch-wise regret formulation and the fact that knowledge of $n$ is only required for the strong $\varepsilon$-exploration result. As discussed in my last question, there might still be room for further improvement, but that is outside the scope of the current paper. I will keep my current score. Good luck :)

---

> > > ### Author Response · Authors · 2026-04-07
> > >
> > > We have incorporated these changes in the updated version of the manuscript. Thank you for the additional feedback!

---

### Official Review · Reviewer_jars · 2026-03-05

**Soundness:** 3
**Presentation:** 3
**Significance:** 2
**Originality:** 2
**Overall Recommendation:** 5
**Confidence:** 3

**Summary:**

The paper investigates a variant of the multi-armed bandit problem in which the set of available arms evolves over time, specifically where the set of active arms changes according to a moving window mechanism. The authors analyze this model under the two classical bandit frameworks: best arm identification and regret minimization. Their study focuses not only on sample complexity and regret, but also on an additional metric, referred to as space complexity, defined as the maximum number of arms that must be stored, a quantity that is specific to this dynamic setting. They derive both upper and lower bounds for each framework, with the best arm identification setting further divided into exact, strong and weak $\epsilon$-exploration regimes. Finally the paper provides experiments to back up the theoretical results.

**Compliance With Llm Reviewing Policy:**

Affirmed.

**Final Justification:**

After reading the rebuttal, I confirm my score.

**Key Questions For Authors:**

1) From the way the setting is presented, it seems that the authors consider a fixed order of the arms among which the window of available ones loops. This makes it so that, after an initial loop the learner knows the arrival sequence. I think this is rather limiting, hence I wonder if this assumption can be relaxed to a more general one where the subset of available arms changes by adding, at each round, a new arm randomly chosen.


2) Do you have any insight about the case in which the $T_j$ are chosen adaptively? What are the issues in adapting the analysis to this case?

**Limitations:**

yes

**Strengths And Weaknesses:**

The paper investigates a novel extension of an existing setting within the broader multi-armed bandit literature. The problem formulation is both interesting and well motivated. The exposition is exceptionally clear, making the paper highly readable and insightful. Finally, the theoretical results are comprehensive and technically sound.

The only weaknesses I identified relate to certain modeling choices, which I discuss in more detail in the “Questions” section.

---

> ### Author Rebuttal · Authors · 2026-03-31
>
> We thank the reviewer for the positive feedback and insightful points. Our responses are as follows.
>
> > From the way the setting is presented, it seems that the authors consider a fixed order of the arms among which the window of available ones loops. This makes it so that, after an initial loop the learner knows the arrival sequence. I think this is rather limiting, hence I wonder if this assumption can be relaxed to a more general one where the subset of available arms changes by adding, at each round, a new arm randomly chosen.
>
> We focus on a stream with a fixed order of arms because our algorithm is designed as a one-pass algorithm (rather than a multi-pass algorithm). Therefore, our model will not admit multiple loops over the arms.
>
> We studied single-pass algorithms specifically because the sliding-window model is motivated by the recent and temporal access to data, and the one-pass setting is more realistic, as it captures the high timeliness of the input stream. That being said, blending the sliding-window model with multi-pass and/or random order streams is a very interesting future direction, where the change of orders will be more natural.
>
> > Do you have any insight about the case in which the T_j are chosen adaptively? What are the issues in adapting the analysis to this case?
>
> Our algorithm does not require the value of $T_j$ until the j-th epoch, i.e., it is actually adaptive. Whether $T_j$ is fixed before start or adaptively chosen by an adversary during execution, the algorithm maintains the same upper bound due to the guarantees for epoch-wise UCB.

---

> > ### Author Rebuttal · Reviewer_jars · 2026-04-02
> >
> > My concern have been adeguately addressed and I maintain my opinion of this being a valid and sound paper.
> > I thank the authors for their rebuttal.

---

> > > ### Author Response · Authors · 2026-04-07
> > >
> > > Thank you for the prompt follow-up. We are glad to hear that your concerns have fully resolved and your overall positive feedback on our work.

---

### Official Review · Reviewer_BjFc · 2026-03-11

**Soundness:** 3
**Presentation:** 3
**Significance:** 2
**Originality:** 4
**Overall Recommendation:** 4
**Confidence:** 3

**Summary:**

This paper proposes the algorithm for sliding window multi armed bandits where the learner must find an approximate best arm among the arms in the given window.

**Compliance With Llm Reviewing Policy:**

Affirmed.

**Final Justification:**

I appreciate the authors' efforts in clarifying these points, and I am maintaining my positive score for this solid and well-supported work.

**Key Questions For Authors:**

Q1. The results presented in Figure 2 illustrate a sharp, non-linear decrease in cumulative regret as the memory size approaches $50$. Could the authors provide a physical or algorithmic explanation for this 'phase transition'? Specifically, does this threshold correspond to a specific ratio between the memory capacity and the number of arms $n$, or is it related to the internal buffer management of the proposed algorithm? Clarifying whether this is a general property of the memory-constrained bandit or a specific artifact of the experimental setup would be highly beneficial."

Q2. While the problem-independent bounds are rigorous, the manuscript lacks a discussion on the fundamental difficulty of deriving instance-dependent (gap-dependent) bounds in this memory-constrained setting. Could the authors elaborate on the specific technical hurdles encountered when attempting to incorporate the mean gap $\Delta$ into the upper and lower bound analysis?

**Limitations:**

Yes.

**Strengths And Weaknesses:**

##  Strengths

### **Technical Soundness**

The paper is rigorous and well-supported. The theoretical derivations are consistent with the empirical results, providing high confidence in the algorithm’s performance and reliability.

### **High-Quality Presentation**

The manuscript is exceptionally well-organized and easy to follow. The authors effectively use motivating examples to ground their work, and the **"Contributions"** section clearly delineates how this approach advances the current state of the art.

### **Practical Originality**

The paper addresses a highly relevant and modern bottleneck: **memory and storage constraints** in large-scale bandit applications. This provides fresh, practical insights that are often overlooked in purely theoretical treatments of the problem.

---

##  Weaknesses

### **Gap in Regret Analysis (Significance)**

While the paper successfully addresses practical memory limitations, the theoretical analysis is currently confined to **problem-independent** bounds.

### **Lack of Instance-Dependency**

The authors do not account for the mean gap $\Delta$ between the arms. As both the upper and lower bounds provided are problem-independent, there remains a potential theoretical gap to be filled.

---

> ### Author Rebuttal · Authors · 2026-03-31
>
> We thank the reviewer for the positive feedback and helpful comments. Our responses are as follows.
>
> > While the paper successfully addresses practical memory limitations, the theoretical analysis is currently confined to problem-independent bounds.
>
> > The authors do not account for the mean gap $\Delta$ between the arms. As both the upper and lower bounds provided are problem-independent, there remains a potential theoretical gap to be filled.
>
> > Q2. While the problem-independent bounds are rigorous, the manuscript lacks a discussion on the fundamental difficulty of deriving instance-dependent (gap-dependent) bounds in this memory-constrained setting. Could the authors elaborate on the specific technical hurdles encountered when attempting to incorporate the mean gap  into the upper and lower bound analysis?
>
> Getting instance-dependent regret bounds is indeed an excellent direction; however, it is not immediately clear how to define instance-dependent bounds for sliding-window algorithms. We remark that since the “best arm” changes over the windows in the sliding-window model, the gaps become harder to define. Furthermore, many natural algorithms for gap-dependent regret bounds in the offline setting can not be directly used in the streaming setting with small memory. Even in the streaming setting, the only work to rigorously establish tight instance-dependent bounds is presented in a very recent paper "Tight Gap-Dependent Memory-Regret Trade-Off for Single-Pass Streaming Stochastic Multi-Armed Bandits" ([COCOON 2025]), where the authors utilized quite novel techniques. As such, we believe that exploring this area is out of the scope of the current work, and it represents a very interesting open problem for future research.
>
> > Q1. The results presented in Figure 2 illustrate a sharp, non-linear decrease in cumulative regret as the memory size approaches 50. Could the authors provide a physical or algorithmic explanation for this 'phase transition'? Specifically, does this threshold correspond to a specific ratio between the memory capacity and the number of arms n, or is it related to the internal buffer management of the proposed algorithm? Clarifying whether this is a general property of the memory-constrained bandit or a specific artifact of the experimental setup would be highly beneficial."
>
> There is a sharp decrease in cumulative regret as the memory size approaches 50, because the memory size approaches the window size $W = 50$ used in the experiment. Additional experiments with varying window sizes, included in the appendix, also demonstrate a similar sharp decline in cumulative regret near their respective window sizes. This is *exactly consistent* with our theoretical results: when $m=o(W)$, regret is high due to expiration issues; when $m=\Omega(W)$, the regret drops sharply.

---

> > ### Author Rebuttal · Reviewer_BjFc · 2026-04-03
> >
> > The authors have provided a comprehensive response to my initial review. My concerns regarding the bounds and experiments are now fully resolved. I appreciate the authors' efforts in clarifying these points, and I am maintaining my positive score for this solid and well-supported work.

---

> > > ### Author Response · Authors · 2026-04-07
> > >
> > > We’re glad our responses addressed your questions, and we thank you for your support; in an updated version of the manuscript, we have added discussion on the instance-dependent bound challenges and highlighted the sharp regret drop near the window size in the experiments.

---

### Official Review · Reviewer_aFim · 2026-03-11

**Soundness:** 3
**Presentation:** 2
**Significance:** 3
**Originality:** 3
**Overall Recommendation:** 5
**Confidence:** 3

**Summary:**

This paper studies a streaming multi-armed bandit problem with arm expiration. In the proposed model, only a window of size W of arms is available at any time. When the algorithm advances to consider a new arm, the arm that falls outside this window immediately expires and can no longer be pulled, regardless of whether it is stored in the algorithm’s memory. The paper investigates several variants of this setting, including pure exploration versus regret minimization objectives and exact versus approximate best-arm identification.

To address these challenges, the authors propose an algorithm based on a bucketization strategy, in which only one representative arm needs to be stored for each bucket. This design significantly reduces memory usage while still enabling the algorithm to identify an approximately optimal arm. The paper provides theoretical guarantees, including upper and lower bounds on sample complexity and regret, and complements the analysis with numerical experiments.

**Compliance With Llm Reviewing Policy:**

Affirmed.

**Final Justification:**

My main concerns have now been addressed. I will maintain my positive rating.

**Key Questions For Authors:**

- Could the authors include additional experiments comparing the proposed method with existing streaming multi-armed bandit algorithms? Although prior algorithms may not be specifically designed for the sliding-window setting, for benchmarking purposes, it may be straightforward to adapt them by forcing the algorithms to discard or ignore expired arms.
- Using pure-exploration algorithms as a basis for regret minimization is often not the most efficient approach. Could the authors elaborate on this design choice? In particular, do the authors expect that a regret-minimization-focused algorithm could potentially achieve better performance for the proposed problem?
- Regarding the notion of epochs, are there any requirements or recommended scaling relationships between the epoch length T_j and the overall horizon T?

**Strengths And Weaknesses:**

Strengths:
- The paper considers several meaningful variants of the problem, including pure exploration versus regret minimization and exact versus approximate formulations, which broadens the scope and relevance of the work.
- The theoretical guarantees are complemented by empirical evaluations on both synthetic datasets and real-world traces, which helps demonstrate the practical performance of the proposed methods.

Weaknesses:
- My main concern is the lack of benchmarking against existing streaming multi-armed bandit algorithms. Including comparisons with prior methods would help better position the proposed approach within the literature and clarify its empirical advantages.
- The presentation could be improved for readability. In particular, the paper introduces a substantial portion of the results before clearly presenting the problem formulation and the algorithmic framework, which makes the overall flow difficult to follow.

---

> ### Author Rebuttal · Authors · 2026-03-31
>
> We thank the reviewer for the positive feedback and insightful questions. Our responses are as follows.
>
> > My main concern is the lack of benchmarking against existing streaming multi-armed bandit algorithms. Including comparisons with prior methods would help better position the proposed approach within the literature and clarify its empirical advantages.
>
> > Could the authors include additional experiments comparing the proposed method with existing streaming multi-armed bandit algorithms? Although prior algorithms may not be specifically designed for the sliding-window setting, for benchmarking purposes, it may be straightforward to adapt them by forcing the algorithms to discard or ignore expired arms.
>
> We want to clarify that our baseline algorithm in the experimental section is already (an adaptation of) an existing streaming algorithm. In fact, both the exploration and regret-minimization baseline algorithms utilize the streaming exploration algorithm. Algorithms designed for the streaming setting cannot be directly used in the sliding-window model due to the expiration of arms, which motivated our adaptations. Our experiments demonstrate that, despite this adaptation, streaming algorithms continue to be affected by expiration.
>
> In the final version of the paper, we will further clarify the baseline algorithms. We will also take advantage of the increased page limits to add more experiments based on baseline streaming algorithms to the main body.
>
> > The presentation could be improved for readability. In particular, the paper introduces a substantial portion of the results before clearly presenting the problem formulation and the algorithmic framework, which makes the overall flow difficult to follow.
>
> Thank you for your valuable comment. While we did mention the informal problem definitions before presenting the results, we acknowledge that our explanation may not have been sufficiently clear. We will enhance this section to improve clarity and better communicate our findings in the revised version.
>
> > Using pure-exploration algorithms as a basis for regret minimization is often not the most efficient approach. Could the authors elaborate on this design choice? In particular, do the authors expect that a regret-minimization-focused algorithm could potentially achieve better performance for the proposed problem?
>
> We apologize for any confusion caused by our presentation and respectfully clarify that our proposed regret-minimization algorithm does not rely on pure exploration. We believe the reviewer is referencing our baseline algorithm, which we employed $\varepsilon$-exploration before committing for comparison purposes. This strategy was chosen because the optimal algorithm in the streaming setting is obtained by $\varepsilon$-exploration (see "Tight regret bounds for single-pass streaming multi-armed bandits" [ICML 2023]). Consequently, our method follows the strategy specified by the optimal streaming algorithm.
>
> > Regarding the notion of epochs, are there any requirements or recommended scaling relationships between the epoch length $T_j$ and the overall horizon $T$?
>
> There are no requirements linking the epoch pulls $T_i$ to the overall horizon $T$. The bound is maximized when all $T_j$ are equal (roughly $O(\sqrt{n W T})$ regret) and minimized when $T_j$ equals $T$ for some $j$ and all other $T_i$ are zero ($O(\sqrt{W T})$ regret).

---

> > ### Author Rebuttal · Reviewer_aFim · 2026-04-04
> >
> > I thank the authors for their thoughtful responses to my questions. I recommend incorporating these clarifications into the main paper to improve its clarity.

---

> > > ### Author Response · Authors · 2026-04-07
> > >
> > > Thank you for your additional feedback. We’re glad our responses addressed your questions. In the updated version of the manuscript, we have incorporated these clarifications, explained that our regret-minimization algorithm does not rely on pure exploration, and provided additional discussion on epoch scaling to improve presentation.

---

### Decision · Program_Chairs · 2026-04-30

**Decision:**

Accept (regular)

**Comment:**

All reviewers reach a consensus in recommending acceptance for this paper. Reviewers agree that this paper has the following strengths:

1. The paper considers several meaningful variants of the sliding-window streaming multi-armed bandit problem, including pure exploration/regret minimization and exact/approximate formulations. This paper is well written and executed.
2. This paper proposes a BUCKET algorithm that is simple and easy to implement, and provides rigorous and complete theoretical results.
3. This paper addresses a relevant problem, i.e., memory and storage constraints, which can be applied to various large-scale decision-making applications.

Reviewers also note limitations: the experiments still have space to improve, such as including more baselines of existing streaming bandit algorithms.

Overall, this paper makes solid contributions to online learning and merits acceptance.